# Requirement of myomaker-mediated stem cell fusion for skeletal muscle hypertrophy

**Qingnian Goh, Douglas P Millay***

Department of Molecular Cardiovascular Biology, Cincinnati Children's Hospital Medical Center, Cincinnati, United States

**Abstract** Fusion of skeletal muscle stem/progenitor cells is required for proper development and regeneration, however the significance of this process during adult muscle hypertrophy has not been explored. In response to muscle overload after synergist ablation in mice, we show that myomaker, a muscle specific membrane protein essential for myoblast fusion, is activated mainly in muscle progenitors and not myofibers. We rendered muscle progenitors fusion-incompetent through genetic deletion of myomaker in muscle stem cells and observed a complete reduction of overload-induced hypertrophy. This blunted hypertrophic response was associated with a reduction in Akt and p70s6k signaling and protein synthesis, suggesting a link between myonuclear accretion and activation of pro-hypertrophic pathways. Furthermore, fusion-incompetent muscle exhibited increased fibrosis after muscle overload, indicating a protective role for normal stem cell activity in reducing myofiber strain associated with hypertrophy. These findings reveal an essential contribution of myomaker-mediated stem cell fusion during physiological adult muscle hypertrophy.

***For correspondence:** douglas.millay@cchmc.org

**Competing interests:** The authors declare that no competing interests exist.

## Introduction

Adult skeletal muscle exhibits a dramatic capacity to regenerate after injury owing to the presence of satellite cells (SCs), the resident muscle stem cells. SCs reside underneath the basal lamina in a quiescent state, and upon injury, become activated to generate a pool of myogenic progenitors (MPs) that differentiate and fuse to each other or existing myofibers to restore structure and function to muscle (*Relaix and Zammit, 2012*; *Yin et al., 2013*). While SCs are clearly necessary for efficient regeneration after acute injury (*Günther et al., 2013*; *Lepper et al., 2011*; *Sambasivan et al., 2011*), the role of SCs during adult muscle hypertrophy and maintenance is much less understood. Moreover, the regulation of myogenic fusion is relatively unknown compared to the knowledge associated with activation, proliferation, and differentiation of SCs. Elucidation of the magnitude of fusion, and the associated molecular regulation of the fusion process, for adult muscle hypertrophy may lead to strategies that augment loss of muscle mass due to chronic disease and aging.

Load-induced hypertrophy is a desired adaptation of exercise resulting in an increase in skeletal muscle mass, thereby enhancing overall size and function of the affected muscles. Fundamentally, skeletal muscle hypertrophy occurs when the rates of muscle protein synthesis exceed those of protein degradation. In addition to activation of protein synthesis pathways directly in the myofiber, a myofiber could hypertrophy through fusion of SCs leading to a greater capacity for transcription of contractile machinery. The contribution of SCs for hypertrophy was initially justified through the concept of the myonuclear domain (*Cheek et al., 1965*), which postulates a coordinated increase between myonuclei number and cytoplasm volume in order to maintain a constant domain size (*Snow, 1990*; *Tamaki et al., 1996*). Moreover, myonuclear accretion precedes an increase in skeletal

**eLife digest** Skeletal muscle has a remarkable capacity to adapt to a variety of stimuli, including an ability to become larger and stronger through exercise. In embryos, new muscles develop from muscle stem cells, which either replicate themselves or "differentiate" into mature muscle cells. Adult muscles also contain stem cells, which are normally dormant, but activate when the muscle is damaged. The stem cells subsequently differentiate and fuse with one another or to existing muscle fibers to restore the muscle. What is not fully understood is whether this fusion process also helps undamaged adult muscles to increase in size (for example, in response to exercise).

Fusion proteins such as myomaker – which specifically acts in muscles – help the stem cells to fuse. To investigate myomaker's role in adult muscle growth, Goh and Millay deleted the gene that produces it from the muscle stem cells of mice. The mice then experienced two weeks of increased muscle activity, after which their muscle growth was compared with that of normal mice that had been subjected to the same activity routine.

Goh and Millay discovered that myomaker is important in muscle stem cells, and not in muscle fibers, for adult muscle growth. After two weeks of increased muscle activity, substantial levels of muscle stem cell fusion had occurred in normal mice, and their muscles had grown significantly. However, the muscles of mice that lacked myomaker in their muscle stem cells did not increase in size.

Additional experiments showed that normal muscle stem cell fusion activates signaling pathways that create new proteins and drive muscle growth. Furthermore, scarring occurred in muscles that lacked myomaker, suggesting that stem cell fusion also protects muscle fibers from damage during increased activity.

Overall, the findings presented by Goh and Millay reveal that the fusion of muscle stem cells is an important event for adult muscle growth. Further studies are now needed to determine the relevance of muscle stem cell fusion during the normal aging process, and to uncover the relationship between fusion and the activation of pro-growth signaling pathways.

muscle size (*Bruusgaard et al., 2010*), and loss of serum response factor (Srf) in the myofiber blunts hypertrophy potentially through a SC-mediated mechanism (*Guerci et al., 2012*), altogether implicating a role for fusion during hypertrophy. Studies utilizing diphtheria toxin (DTA)-mediated ablation of SCs have revealed that effective myofiber hypertrophy occurs in the absence of satellite cells and that SCs are not required to maintain muscle mass during aging (*Fry et al., 2015*; *McCarthy et al., 2011*), however SC ablation leads to impaired coordination and diminished muscle performance (*Jackson et al., 2015*). Additionally, SCs contribute to myofibers during aging of sedentary mice although SC activity was not globally required to maintain muscle size (*Keefe et al., 2015*; *Pawlikowski et al., 2015*). Recently, a report utilizing a similar DTA SC ablation model indicates that effective myofiber hypertrophy does not occur in the absence of SCs (*Egner et al., 2016*). Thus, there is no consensus regarding the contributions of SCs during adult skeletal muscle hypertrophy, highlighting the need for more independent models that assess the biology of SC activity.

Myomaker (*Tmem8c*) is a muscle-specific membrane protein necessary for fusion of embryonic and adult MPs (*Millay et al., 2013*). In adult muscle, myomaker expression is undetectable but rapidly induced in SC descendants after cardiotoxin (CTX) injury. In this acute injury setting where there is complete destruction of the muscle architecture, the ensuing restoration is the sole responsibility of SCs. Here, all MPs exhibit upregulation of myomaker, and we have shown that fusion is more efficient if both fusing cells express myomaker (*Millay et al., 2016*). The regulation and function of myomaker in the SC or myofiber compartment in an environment where myofibers are maintained and not destroyed, such as through increased workload and exercise, is unknown.

To investigate the role of myomaker-mediated fusion in skeletal muscle hypertrophy, we explored the expression of myomaker on SCs and myofibers, and generated fusion-incompetent SCs through targeted deletion of myomaker. We show that myomaker is upregulated in SCs upon muscle overload (MOV), but significant activation of myomaker within the myofiber is not detected. In the absence of myomaker induction in SCs, we report an impaired hypertrophic response to MOV, which

is associated with a failure to activate Akt/mTOR-mediated protein synthesis within the myofiber. Collectively, our findings demonstrate that effective SC fusion is required for optimal muscle hypertrophy following a load-induced stimulus.

## Results

### Dynamic expression of myomaker during muscle overload coincides with fusion

To determine the regulation of MP fusion during adult skeletal muscle hypertrophy, we first assessed the temporal kinetics of myomaker expression after MOV. In wild-type (WT) mice at baseline, myomaker mRNA and protein are not readily detected in muscle, however myomaker is acutely induced after MOV (*Figure 1A and B*). Expression of myomaker peaked 7 days after MOV, and was downregulated at later time points. To elucidate if myomaker was induced in activated MPs or by the myofiber during MOV, we analyzed *Tmem8c*$^{LacZ/+}$ mice, which contain a *LacZ* cassette in intron 1 of the myomaker locus (*Millay et al., 2014*). This allele results in exon 1 of myomaker splicing with *LacZ*, generating a transcript with exon 1 of myomaker, and an internal ribosome entry site (IRES) that allows independent translation of *LacZ*. Thus, in this system *LacZ* serves as a readout for myomaker transcription but not myomaker localization. X-gal staining of the plantaris muscle from *Tmem8c*$^{LacZ/+}$ mice at multiple time points after MOV revealed detectable *LacZ* in discrete locations surrounding the myofiber at day 3 of MOV, and in myofibers at later stages of MOV (*Figure 1C*). To understand the myogenic state of the different types of LacZ$^+$ cells we stained serial sections with either x-gal or embryonic myosin (myh3), a marker of muscle differentiation. Here we observed a population of large LacZ$^+$ cells and a population of small LacZ$^+$ cells that were in the presumptive SC position (*Figure 1—figure supplement 1*). We classified the large LacZ$^+$ cells as myofibers due to their size, and these were either myh3$^+$ or myh3$^-$. For myofibers, LacZ$^+$ myh3$^+$ cells exhibited stronger x-gal staining compared to the LacZ$^+$ myh3$^-$ cells (punctate staining) suggesting that the former population are de novo fibers formed from the fusion of MPs (*Figure 1—figure supplement 1*). We interpret the punctate LacZ$^+$ myh3$^-$ myofibers as existing fibers that have fused with a MP. The population of small LacZ$^+$ cells were either myh3$^+$ (differentiated myocytes) or myh3$^-$ (MPs), and we classified these as the non-myofiber population (*Figure 1—figure supplement 1*). Quantification of these populations multiple days after MOV revealed an increase in myomaker-expressing MPs at the early stages of MOV (days 3 and 7), followed by a reduction at later stages of MOV (days 10 and 14) (*Figure 1C*). In contrast, the majority of LacZ$^+$ myofibers were observed at later stages of MOV, with negligible occurrence at 3 days after MOV. Thus, myomaker exhibits a contrasting expression pattern in MPs compared to the myofiber compartment in response to a load-induced stimulus.

To determine if the kinetics of myomaker expression temporally coincides with fusion of MPs during MOV, we examined fusion through 5-bromo-2'-deoxyuridine (BrdU) labeling of proliferating cells. After synergist ablation surgery, WT mice were treated with BrdU either during the initial or final seven days of the overload stimulus, and fusion was defined as the incorporation of a BrdU$^+$ nucleus within a dystrophin$^+$ myofiber. Labeling of fusion-competent MPs during the first 7 days of MOV resulted in a greater percentage of myofibers containing a BrdU$^+$ nucleus compared to labeling during the final 7 days of MOV (*Figure 1D*). While these data indicate that fusion-competent MPs are generated throughout the MOV stimulus, there is greater labeling of fusogenic cells during the initial stages of MOV when myomaker-LacZ$^+$ MPs are more prominent, suggesting a temporal relationship between myomaker expression and fusion.

### Myomaker is activated primarily in muscle progenitors

While MP expression of myomaker-LacZ indicates that myomaker is actively transcribed in progenitor cells, LacZ activity in myofibers could result from fusion with a LacZ$^+$ MP or direct transcription of myomaker-LacZ from a myonucleus. To determine the source of LacZ in myofibers during MOV, we blocked the ability of MPs to fuse through genetic deletion of myomaker. *Tmem8c*$^{LacZ/loxP}$ mice were bred with satellite cell-specific Cre recombinase (*Pax7*$^{CreERT2/+}$) mice (*Lepper et al., 2009*), and subsequently treated with tamoxifen to genetically delete myomaker in muscle progenitor cells (myomaker scKO). Controls include tamoxifen-treated *Tmem8c*$^{+/+}$; *Pax7*$^{CreERT2/+}$ mice, or vehicle-treated *Tmem8c*$^{LacZ/loxP}$; *Pax7*$^{CreERT2/+}$ mice (*Figure 2—figure supplement 1*). Mice were injected

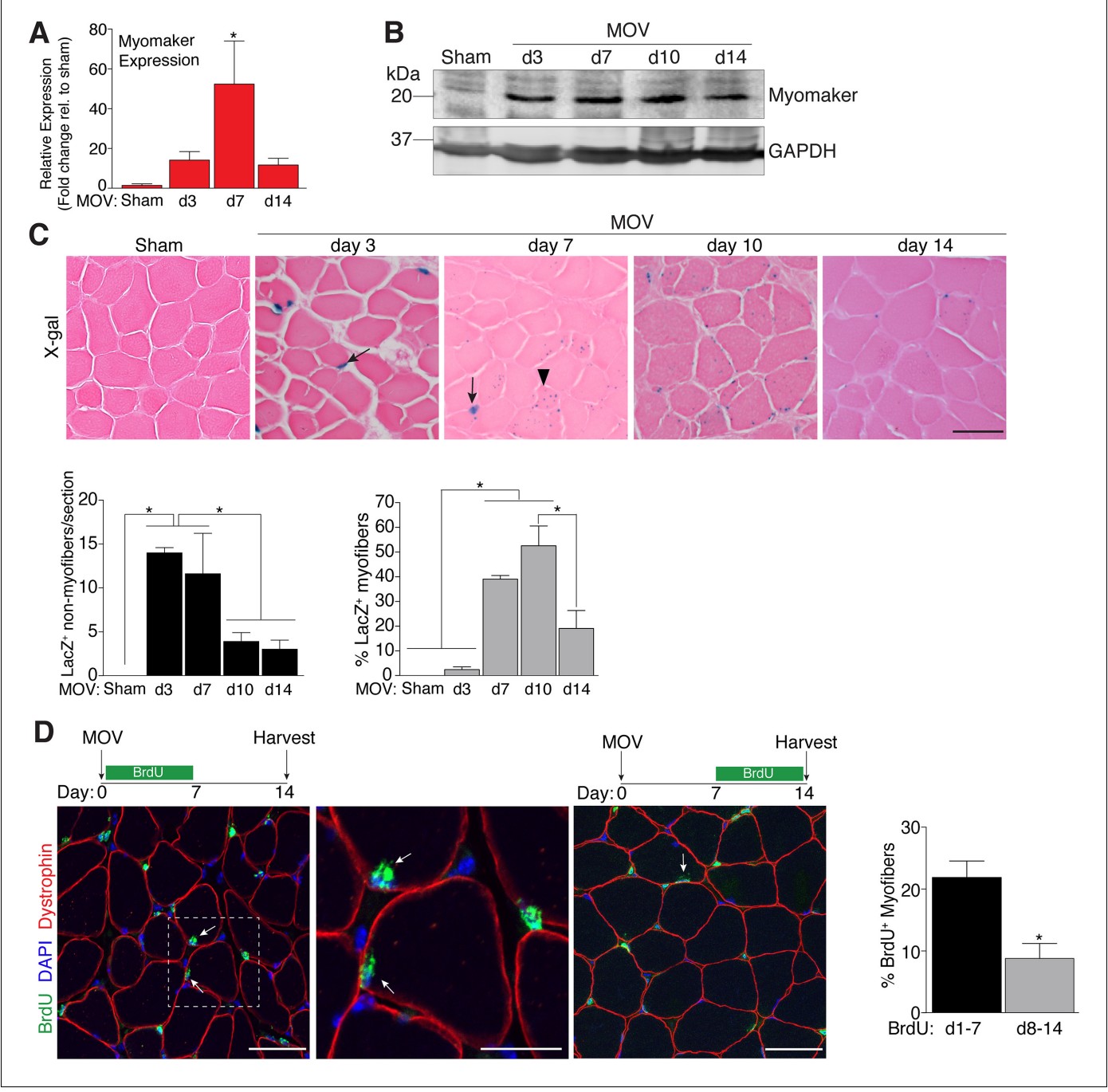

**Figure 1.** Regulation of myomaker activation and fusion during load-induced hypertrophy. Myomaker expression at various time points after MOV was assessed by qPCR (A), and western blot analysis (B), showing induction of myomaker at all stages of MOV (n = 2–4 mice). (C) Tmem8c$^{LacZ/+}$ mice were subjected to MOV and plantaris sections were X-gal stained at multiple time points after surgery. LacZ, a surrogate for myomaker expression, was observed in MPs (arrows) during the early stages of MOV, and in myofibers (arrowheads) in the later stages of MOV. Quantification of the number of LacZ$^+$ non-myofibers indicates myomaker is robustly activated in MPs at day 3 and day 7 of MOV but the number is reduced at day 10 and day 14 (n = 3–5 mice). Quantification of LacZ$^+$ myofibers demonstrates the majority of expression occurs at day 7 and day 10 after MOV (n = 2500–4,100 myofibers from 3–5 independent mice). (D) Fusion of MPs with myofibers was assessed by labeling proliferating cells with BrdU and tracking their incorporation into a myofiber, identified by immunostaining with a dystrophin antibody. Mice were subjected to MOV and treated with BrdU during the initial 7 days or the last 7 days after MOV. Fusion was scored as a BrdU$^+$ nucleus within a dystrophin$^+$ myofiber as depicted by the arrows. Quantification of the percentage of myofibers containing a BrdU$^+$ nucleus shows an increased labeling of fusion competent satellite cells during the first 7 days of MOV, which correlates with the highest expression of myomaker in satellite cells (n = 380–1,511 myofibers from 3–9 independent mice). Data are represented as mean ± SEM, *p<0.05. Scale bars: 50 µm, except inset in (D) which represents 25 µm.

*Figure 1 continued on next page*

*Figure 1 continued*

The following figure supplement is available for figure 1:

**Figure supplement 1.** Day 7 MOV serial sections were stained with x-gal or immunostained with laminin and embryonic myosin (myh3) antibodies.

with tamoxifen for 5 consecutive days prior to MOV and maintained on a tamoxifen dosage throughout the protocol. qPCR analysis confirmed the efficiency of this tamoxifen-based strategy, as myomaker expression was eliminated in myomaker scKO muscle compared to control muscle 7 days after MOV (*Figure 2A*). To be certain that this model maintains the presence of SCs during MOV we crossed the *Rosa26*$^{mTmG}$ genetic lineage reporter with our *Tmem8c*$^{LacZ/loxP}$; *Pax7*$^{CreERT2/+}$ mice (*Figure 2—figure supplement 1*). This results in a ubiquitously expressed membrane-targeted tdTomato (mTom), but upon Cre-mediated recombination tdTomato is excised allowing membrane-GFP expression. These mice were subjected to MOV and assayed for the presence of mGFP$^+$ cells (MPs) using flow cytometry. We observed a non-significant reduction in mGFP$^+$ cells after MOV in myomaker scKO muscle (*Figure 2—figure supplement 2*). We also performed immunostaining for Pax7, which revealed no major differences in SC number during MOV when normalized to fiber number (*Figure 2—figure supplement 2*). These results indicate that while there could be subtle alterations in SC dynamics, the myomaker scKO model does not significantly alter the frequency of muscle stem cells.

We next assessed how the loss of myomaker in MPs impacts LacZ$^+$ activity within the muscle. X-gal staining of plantaris sections from myomaker scKO mice after MOV demonstrates that myomaker expression is restricted to non-myofibers (*Figure 2B*). Quantification of LacZ$^+$ non-myofibers revealed an accumulation at 7 days post-MOV in myomaker scKO mice compared to control mice, reflecting the inability of myomaker-deficient MPs to fuse to myofibers (*Figure 2B*). We also observed a lack of LacZ activity in myofibers in myomaker scKO mice at days 7–14 of MOV, suggesting that LacZ in myofibers in control mice is contributed through fusion of MPs (*Figure 2B*). Overall, these data indicate that myomaker is not actively transcribed from a myofiber nucleus in response to MOV.

One possibility for the lack of myonuclear transcription in myomaker scKO mice is that the myofiber requires expression of myomaker on MPs to activate myomaker transcription within the myofiber. To exclude this possibility, we genetically deleted myomaker using a myofiber-specific Cre and assessed myomaker levels after MOV. Specifically, we crossed *Tmem8c*$^{LacZ/loxP}$ mice with a tamoxifen-inducible Cre under control of the human skeletal actin promoter (*ACTA1*$^{CreERT2}$). We treated *Tmem8c*$^{LacZ/loxP}$; *ACTA1*$^{CreERT2}$ mice with tamoxifen (myomaker mKO) for 5 days prior to MOV and analyzed myomaker transcript levels through qPCR analysis. If myomaker was transcribed from a myofiber nucleus in the presence of WT MPs, we would expect a reduction of myomaker levels in myomaker mKO muscle after MOV. However, myomaker mKO MOV plantaris did not exhibit a reduction in myomaker levels compared to control MOV, whereas a reduction was achieved in myomaker scKO MOV plantaris (*Figure 2C*).

To be certain the *ACTA1*$^{CreERT2}$ was functional, we assessed myomaker levels in myomaker mKO mice when tamoxifen delivery was maintained after MOV. Using this protocol, we did detect a reduction of myomaker expression in myomaker mKO mice, suggesting a functional *ACTA1*$^{CreERT2}$ allele (*Figure 2—figure supplement 3*). Myomaker deletion in myomaker mKO mice was achieved only with maintenance of tamoxifen after MOV, likely because the human skeletal actin (*ACTA1*) promoter is activated as MPs differentiate and prior to fusion. Taken together, our results suggest the LacZ expression observed in control myofibers is acquired primarily from LacZ$^+$ MPs through fusion, and not from transcription by existing myofiber nuclei. Hence, myomaker is activated mainly in MPs during muscle overload.

## Satellite cell-derived myomaker is required for overload-induced muscle hypertrophy

Since myomaker is mainly activated in MPs during MOV, we utilized myomaker scKO mice to determine the role of myomaker during hypertrophy. To ensure effective deletion of myomaker, control

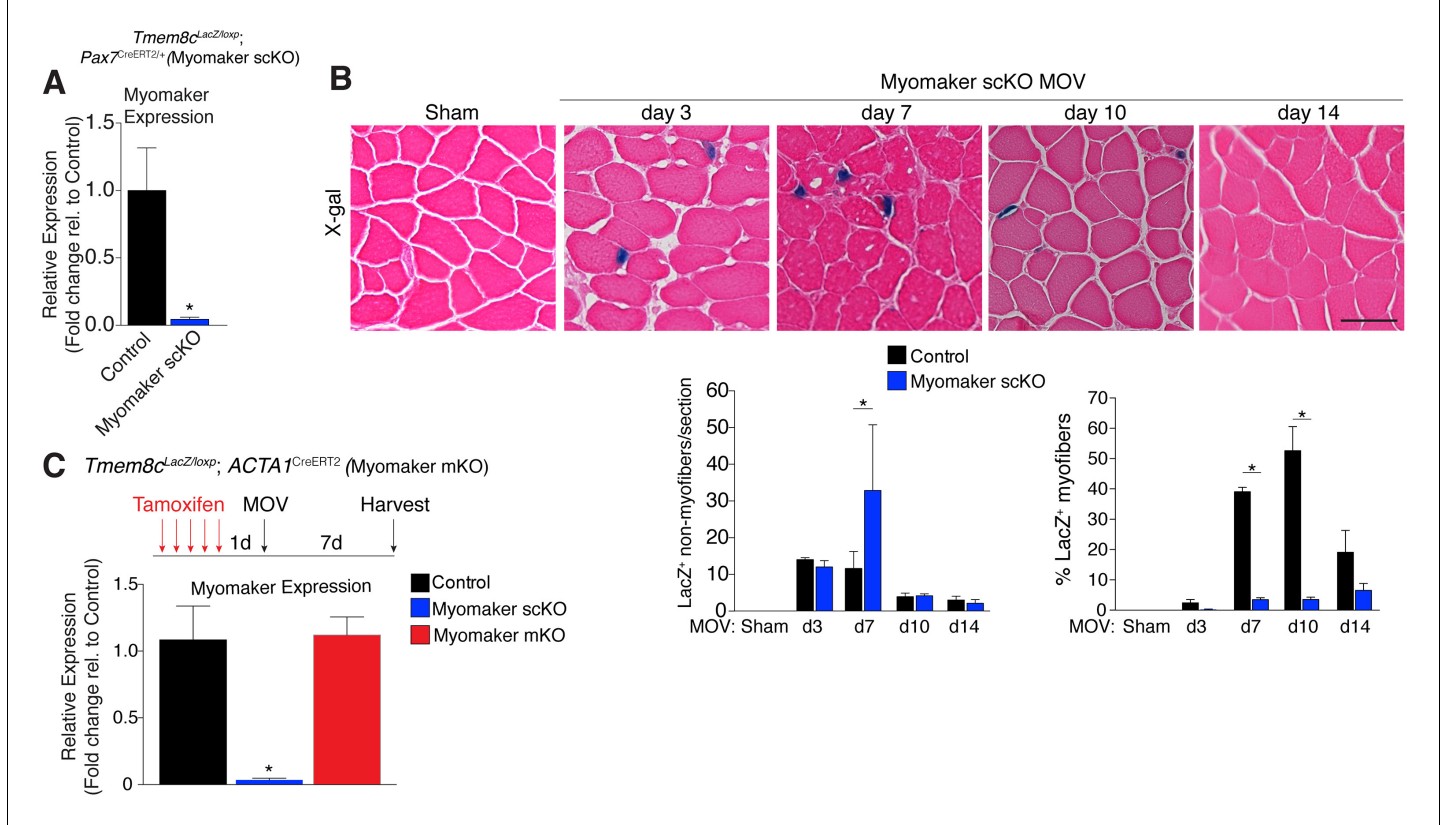

**Figure 2.** Myomaker is mainly expressed in MPs and not myofibers during MOV. (**A**) *Tmem8c^LacZ/loxP^; Pax7^CreERT2/+^* mice (myomaker scKO) were treated with tamoxifen to ablate myomaker specifically in SCs. Control mice include vehicle-treated *Tmem8c^LacZ/loxP^; Pax7^CreERT2/+^* mice and tamoxifen-treated *Tmem8c^+/+^; Pax7^CreERT2/+^* mice. Mice were subjected to MOV and myomaker expression was assessed through qPCR analysis 7 days post-surgery showing that myomaker was efficiently deleted in myomaker scKO muscle (n = 3–4 independent mice). (**B**) To determine the origin of LacZ staining observed in control myofibers (*Figure 1C*), MOV was performed on myomaker scKO mice and sections were X-gal stained. Ablation of myomaker in SCs results in restriction of myomaker expression to MPs and lack of expression in the myofiber. Quantification of LacZ+ non-myofibers shows a significant increase at day 7 in myomaker scKO mice compared to control mice because myomaker scKO are fusion defective and thus remain outside the myofiber (n = 3–5 mice). The percentage of LacZ+ myofibers are reduced in *myomaker^scKO^* mice suggesting that myofiber LacZ is acquired through fusion of MPs (n = 2500–4,100 myofibers from 3–5 independent mice). In the quantification of (**B**), the control bars are from *Figure 1C*. (**C**) To genetically assess the requirement of myofiber-derived myomaker, tamoxifen was administered to *Tmem8c^LacZ/loxP^; ACTA1^CreERT2^* mice (myomaker mKO) prior to MOV. qPCR analysis of myomaker in control, myomaker scKO, and myomaker mKO mice after MOV shows that myomaker is only reduced in myomaker scKO mice, demonstrating that myomaker is not transcribed from a myofiber nucleus (n = 3–4 mice). Data are represented as mean ± SEM, *p<0.05. Scale bar: 50 μm.

The following figure supplements are available for figure 2:

**Figure supplement 1.** Schematic showing the mice used to ablate myomaker in satellite cells, and their associated control groups.

**Figure supplement 2.** Myogenic progenitors are maintained in myomaker scKO muscle during MOV.

**Figure supplement 3.** Myomaker expression is reduced in myomaker mKO mice when tamoxifen is present concomitant with satellite cell activation suggesting *ACTA1^CreERT2^* is active in myoblasts prior to fusion (n = 3–4 mice).

and myomaker scKO mice were subjected to five daily doses of tamoxifen prior to MOV, and kept on a tamoxifen regimen until sacrifice (*Figure 3A*). Following 14 days of overload, evaluation of plantaris weight normalized to tibia length revealed a two-fold increase in control MOV compared to sham (*Figure 3B*). In contrast, myomaker scKO MOV plantaris was smaller than control MOV, although still increased above myomaker scKO sham weight (*Figure 3B*). Muscle responds to a load-induced stimulus by generating new myofibers and activating hypertrophic pathways in existing

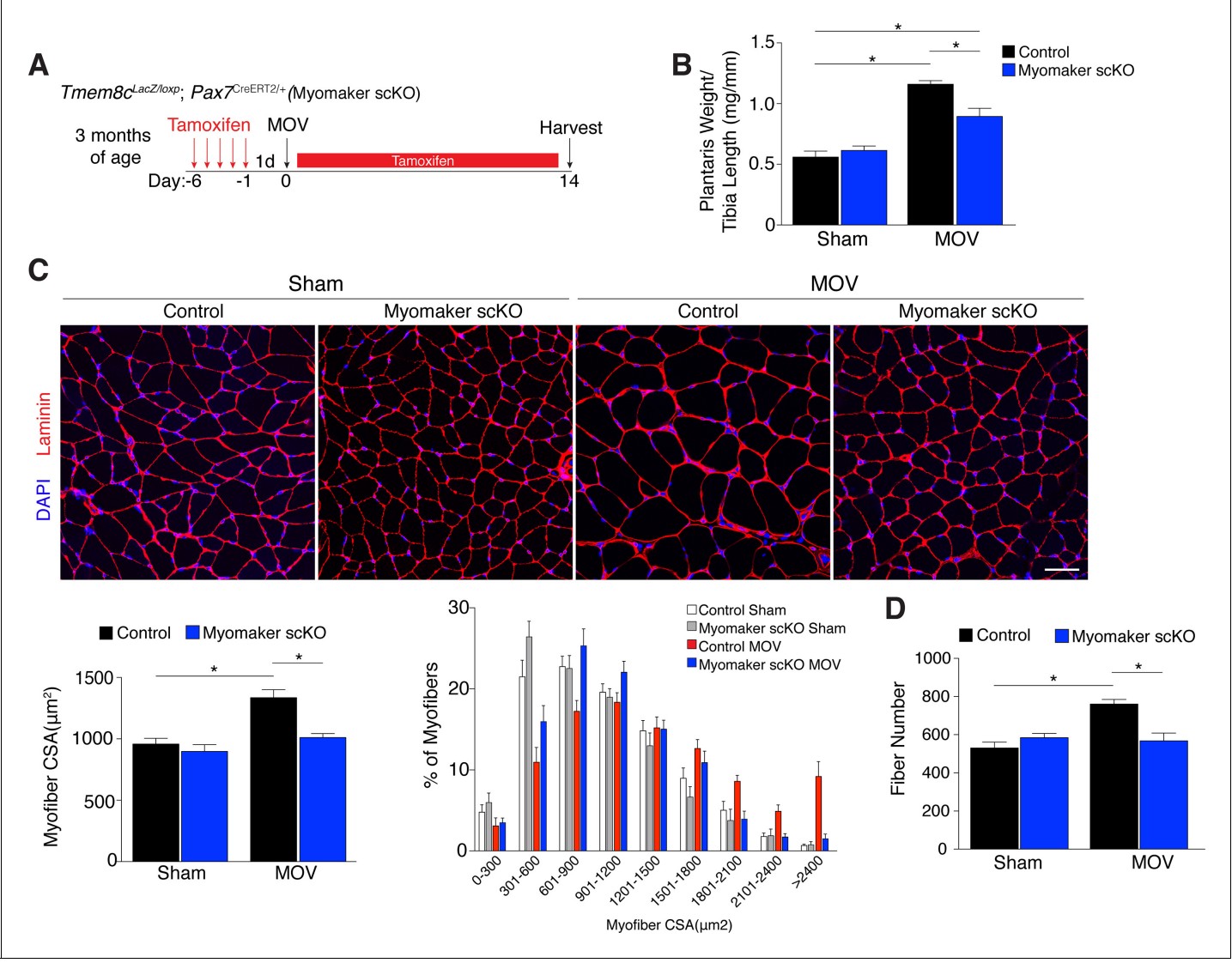

**Figure 3.** Genetic loss of myomaker in MPs results in an absence of muscle hypertrophy. (**A**) Schematic showing timing of tamoxifen treatment by intraperitoneal (IP) injections, MOV, and tissue harvest. Tamoxifen was maintained after MOV through IP injections every other day. (**B**) Plantaris weight normalized to tibia length indicates a dramatic increase in size in control mice, which is blunted in myomaker scKO mice (n = 7–13 mice). (**C**) Representative laminin-stained plantaris sections shows an increase in myofiber size after muscle overload in control mice but not in myomaker scKO mice. Quantification of cross-sectional area (CSA) of myofibers at the plantaris mid-belly reveals a significant inhibition of myofiber hypertrophy with satellite cell ablation of myomaker (n = 3200–5,400 myofibers from 7–11 independent mice). Relative frequency of myofiber sizes in the various groups of mice demonstrates that control muscle exhibits a greater percentage of larger fibers after MOV. (**D**) Fiber number was quantitated at the mid-belly of the plantaris, and reveals a significant inhibition of new fiber formation in plantaris muscle when myomaker was ablated in satellite cells (n = 7–11 mice). Data are represented as mean ± SEM, *p<0.05. Scale bar: 50 μm.

The following figure supplement is available for figure 3:

**Figure supplement 1.** All fiber types fail to hypertrophy in myomaker scKO muscle after MOV.

myofibers. Indeed, through histological examination of the plantaris after MOV, in control mice we observed an increase in myofiber size, a more direct evaluation of hypertrophy (*Figure 3C*). This characteristic was not observed in myomaker scKO MOV plantaris sections, highlighting the importance of myomaker on MPs for efficient hypertrophy (*Figure 3C*). Cross-sectional area (CSA) analysis of fiber size revealed an increase in control MOV compared to control sham (*Figure 3C*). In contrast,

no differences were detected between myomaker scKO sham and MOV, indicating defective myo-fiber hypertrophy following MOV with genetic deletion of myomaker in MPs (*Figure 3C*). In addition, frequency distribution analysis confirms the presence of fewer large myofibers in myomaker scKO muscles after overload (*Figure 3C*). Given that skeletal muscle is comprised of multiple different fiber types that have the capacity to dynamically adapt in response to physiological stimuli, we assessed if all fiber types were resistant to hypertrophy after loss of myomaker in SCs. We immunos-tained sections after 14 days of MOV with antibodies specific to Type I, Type IIa, and Type IIb myo-sin and quantified myofiber CSA for each fiber type. This analysis revealed no differences in fiber type frequency after MOV, and that genetic ablation of myomaker in SCs resulted in a lack of hyper-trophy of Type IIa, IIb, and IIx myofibers (*Figure 3—figure supplement 1*).

Furthermore, the total number of fibers is elevated in control MOV compared to control sham mice, but fiber number is similar between sham and overloaded muscles in myomaker scKO mice (*Figure 3D*). These data suggest an inability of myomaker-deficient MPs to fuse to each other to generate de novo myofibers, indicating a distinct requirement for myomaker in augmenting fiber number during the response to overload. The blunted hypertrophic phenotype observed in myo-maker scKO mice can therefore be attributed to an impairment in hypertrophy of existing myofibers and a failure to generate de novo fibers following overload. Collectively, our findings demonstrate the requirement of satellite cell-derived myomaker for load-induced adult skeletal muscle hypertrophy.

## Fusion during MOV is blocked through deletion of myomaker in SCs

We utilized two independent approaches to determine whether the diminished hypertrophic response to overload in myomaker scKO mice is associated with deficits in MP fusion. First, mice were administered daily dosages of BrdU post-surgery throughout the entire duration of MOV while maintaining the tamoxifen regimen (*Figure 4A*). As described above, BrdU labels proliferating cells (including MPs) and fusion was assayed by tracking the incorporation of a BrdU$^+$ nucleus into a dystrophin$^+$ myofiber. After 14 days of overload in control mice, BrdU$^+$ nuclei within a dystrophin-stained myofiber were readily detected (*Figure 4B*). In comparison, BrdU$^+$ nuclei reside primarily in the interstitium between myofibers in overloaded myomaker scKO mice, indicating efficient abro-gation of fusion (*Figure 4B*). Indeed, quantification of the percentage of BrdU$^+$ myofibers revealed ~30% of myofibers in control mice underwent a fusion event, however in myomaker scKO mice BrdU$^+$ myofibers are at basal levels observed in shams (*Figure 4B*). Given that this is one level of a muscle, if a BrdU myonucleus was above or below this plane then that fiber would not be counted as a positive fusion event. Therefore, this BrdU analysis may underestimate the extent of fusion that occurs during MOV.

To complement our BrdU assay and establish greater insight into the magnitude of fusion during physiological hypertrophy, we used a fluorescent genetic reporter that offers enhanced sensitivity in detecting fusion events. Specifically, $Tmem8c^{+/+}$; $Pax7^{CreERT2/+}$; $Rosa26^{mTmG}$ mice (controls) and $Tmem8c^{LacZ/loxP}$; $Pax7^{CreERT2/+}$; $Rosa26^{mTmG}$ (myomaker scKO) mice were treated with tamoxifen to induce expression of mGFP in MPs (*Figure 4C*). With this strategy, existing myofibers will maintain mTom expression and also be mGFP$^+$ if they have fused with MPs, while myofibers that are solely mGFP$^+$ would represent de novo myofibers arising from the fusion of MPs with each other. After muscle overload the majority of existing myofibers in control mice undergo fusion with MPs, evalu-ated by expression of both mTom and mGFP, which was not observed in overloaded myofibers of myomaker scKO mice (*Figure 4D*). To quantify fusion in this genetic model, we isolated single myo-fibers and analyzed their relative mGFP intensity normalized to total fiber area using ImageJ (*Figure 4E*). Here, we detected a significant increase in mGFP intensity of single fibers in control mice after overload, which is reduced to sham levels after genetic ablation of myomaker in MPs (*Figure 4E*). Our overall findings substantiate that expansive MP fusion occurs during load-induced skeletal muscle hypertrophy, and that myomaker expression on MPs is essential for this myogenic process. Moreover, these results indicate that myomaker-mediated fusion is necessary for myofiber hypertrophy in response to physiological stimuli.

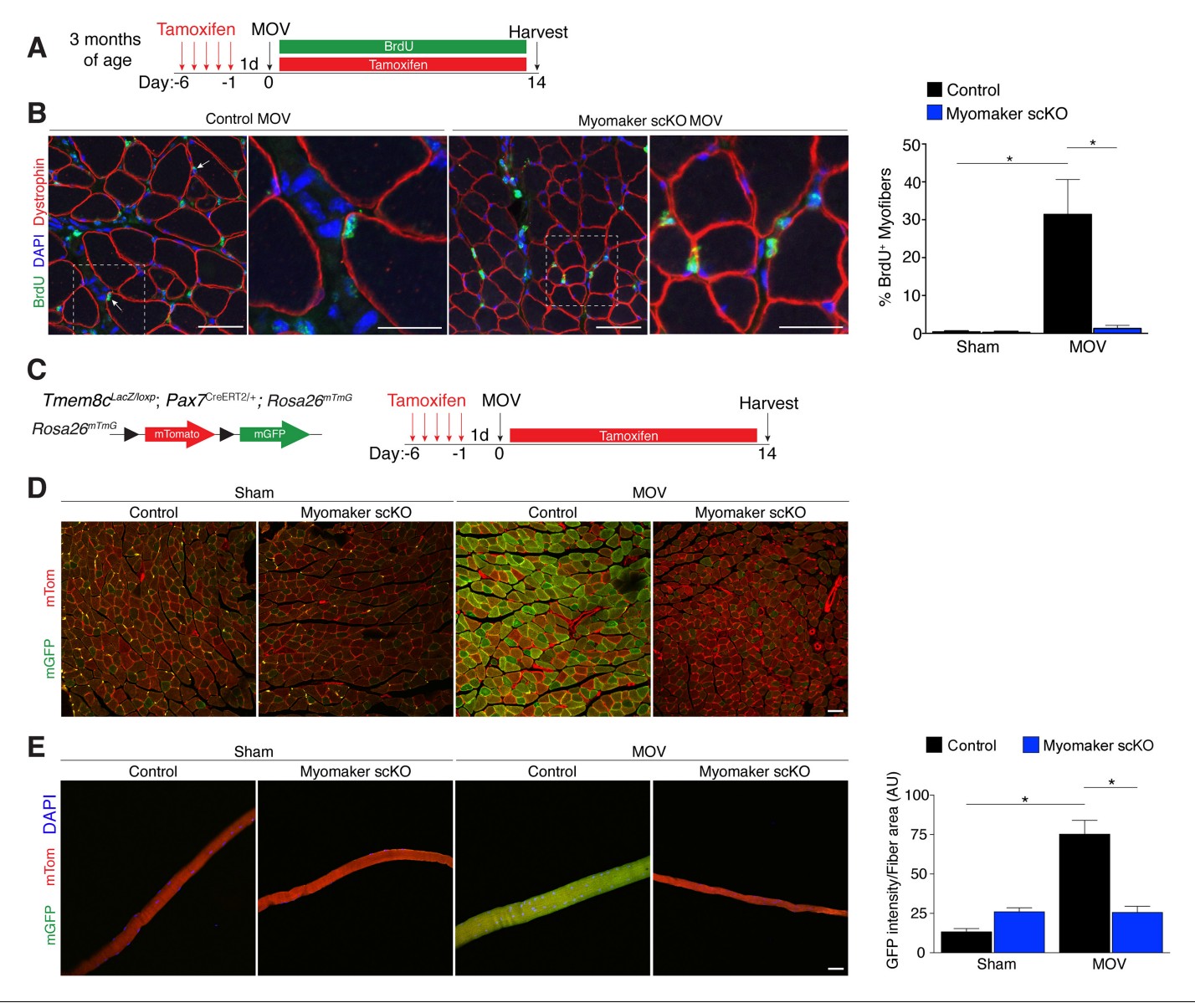

**Figure 4.** Myomaker is required for fusion during overload-induced muscle hypertrophy. (**A**) Schematic showing treatment with tamoxifen and BrdU. (**B**) Representative images of BrdU+ nuclei within a dystrophin-stained myofiber (arrows) indicating fusion. In myomaker scKO mice, BrdU+ nuclei were observed only in the interstitium between myofibers. Quantification of the percentage of BrdU+ myofibers shows that 30% of fibers undergo fusion in control samples after MOV while this is dramatically reduced in myomaker scKO mice (n > 300 fibers from three independent mice). (**C**) $Tmem8c^{LacZ/loxP}$; $Pax7^{CreERT2/+}$ were crossed with $Rosa^{mTmG}$ mice to genetically assay fusion. Tamoxifen-treated $Tmem8c^{+/+}$; $Pax7^{CreERT2/+}$; $Rosa^{mTmG}$ mice were used as controls. Upon tamoxifen treatment, SCs will be converted from mTomato-expressing to mGFP-expressing. mGFP+ myofibers indicates fusion with MPs. (**D**) In sham mice, only minimal mGFP+ fibers were observed. Control MOV mice display numerous mGFP+ myofibers, indicating the majority of myofibers undergo a fusion event. Myomaker scKO mice exhibited marginal mGFP+ fibers (n = 4 mice). (**E**) Single fibers were isolated from control and myomaker scKO mice and analyzed for mTomato and mGFP expression. Quantification of mGFP intensity in single fibers shows an increase in control MOV, which is inhibited when myomaker is ablated in satellite cells (n = 90–120 myofibers from 3–4 independent mice). Data are represented as mean ± SEM, *p<0.05. Scale bars: 50 μm, except insets in (**B**) which are 25 μm.

## Altered muscle signaling and fibrosis in fusion-incompetent muscle during hypertrophy

An established mechanism for eliciting muscle hypertrophy is through the Akt1 pathway, which activates mTOR and p70s6k and ultimately leads to enhanced protein synthesis (*Egerman and Glass,*

*2014*). We explored how this pathway is perturbed with defective MP fusion during adult muscle hypertrophy. To this end, western blot analysis revealed an expected increase in phosphorylated levels of Akt (Ser473) and p70s6k (Thr389) in control MOV muscles at 7 and 14 days of overload (*Figure 5A*). Activation of these pathways was significantly blunted in myomaker scKO overload muscle, highlighting a mechanism by which myomaker expression on MPs regulates hypertrophy

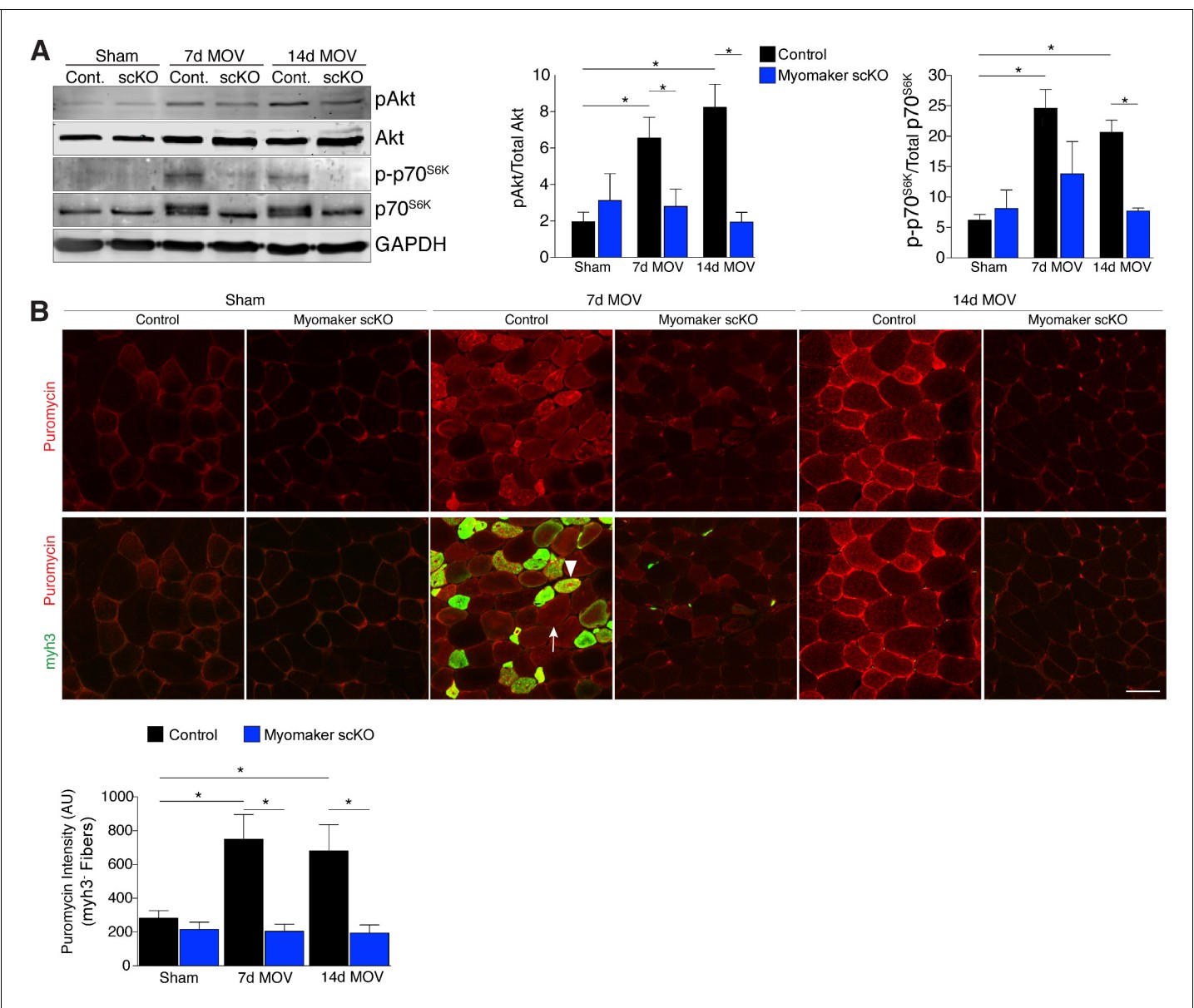

**Figure 5.** Hypertrophic signaling pathways are reduced in the absence of fusion during MOV. (**A**) Western analysis for pAkt (Ser473), total Akt, p-p70$^{S6K}$ (Thr389), and total p70$^{S6K}$. Control plantaris muscle displays pAkt and p-p70$^{S6K}$ activation after MOV compared to sham controls, however the pathways are not fully activated in myomaker scKO MOV muscles. Quantification of the western signal obtained for pAkt normalized to total protein, and quantification of the western signal obtained for p-p70$^{S6K}$ normalized to total protein both reveal altered Akt/mTOR signaling in myomaker scKO muscles after MOV (n = 4 mice). (**B**) Analysis of protein synthesis through puromycin incorporation into nascent peptides. Mice were injected with puromycin and sacrificed thirty minutes later. Sections were then immunostained with antibodies to puromycin and myh3, a marker of newly generated muscle cells. Puromycin staining is increased in both myh3$^{+}$ myofibers (de novo, arrowhead) and myh3$^{-}$ myofibers (existing, arrow) in control samples after MOV. Only myh3$^{-}$ myofibers were observed at 14 days of MOV. Quantification revealed significantly more puromycin incorporation in myh3$^{-}$ myofibers in control samples at both 7 and 14 days after MOV compared to myomaker scKO MOV muscles (2200–3400 fibers from n = 3–4 independent mice). Data are represented as mean ± SEM, *p<0.05. Scale bar: 50 μm.

(*Figure 5A*). Additionally, total levels of p70s6k were elevated in control MOV muscles compared to myomaker scKO overload muscles, which could be indicative of enhanced overall protein translation in control MOV samples during hypertrophy (*Bakker et al., 2016*). Quantitative analysis of normalized pAkt and p-p70s6k levels further demonstrates significant disruptions in the Akt/mTOR pathway in fusion-incompetent muscles subjected to overload (*Figure 5A*).

The ultimate function of mTOR signaling is to activate protein synthesis in myofibers for hypertrophy, therefore we directly investigated rates of protein synthesis after MOV using a technique by which puromycin is incorporated into nascent polypeptides, and is thus a readout of active protein synthesis (*Schmidt et al., 2009*). We injected puromycin into control and myomaker scKO mice thirty minutes prior to sacrifice and assayed for puromycin incorporation into myofibers through immunostaining of muscle sections with a puromycin antibody (*Goodman et al., 2011*). We also co-immunostained these sections with embryonic myosin (myh3) to evaluate the relative contribution of protein synthesis from de novo and existing myofibers. We observed minimal puromycin in sham samples, however in control muscle puromycin was elevated in both myh3$^-$ (existing) and myh3$^+$ (de novo) myofibers 7 days after MOV (*Figure 5B*). At 14 days of MOV, we detected minimal myh3$^+$ myofibers although puromycin remained elevated in existing fibers of control mice, whereas myomaker scKO muscle did not exhibit significant puromycin incorporation at either time point analyzed (*Figure 5B*). Of note, we did not observe myh3$^+$ cells in myomaker scKO muscle 7 days after MOV, further demonstrating that myomaker is required to form new myofibers (*Figure 5B*). Given that myh3$^+$ myofibers are not present in myomaker scKO muscle we only quantified the puromycin intensity in myh3$^-$ myofibers, which revealed a significant inability for fusion-incompetent muscle to activate protein synthesis (*Figure 5B*). Our findings indicate that SC-derived myomaker is required for full activation of Akt/mTOR signaling and protein synthesis in the myofiber, which could serve as a potential mechanism though which fusion drives the hypertrophic response.

In addition to the activation of signaling pathways to achieve optimal hypertrophy, muscle hypertrophy is also associated with remodeling of extracellular matrix (ECM) content. While the mechanisms that govern ECM alterations during muscle hypertrophy and regeneration are relatively unknown, one aspect of this regulation is dependent on the presence of MPs (*Fry et al., 2014*, *2015*; *Murphy et al., 2011*). Thus, we sought to determine the consequence on ECM in fusion-defective muscle that is sensing increased load. We performed Masson's Trichrome staining on control and myomaker scKO sham and MOV muscle, which revealed modest fibrosis in control MOV, indicating that ECM remodeling is a normal process for hypertrophy (*Figure 6*). In contrast, we observed an increase in fibrosis in the overloaded muscles of myomaker scKO mice (*Figure 6*). Quantification of the fibrotic area indeed revealed a significant increase in fibrosis in myomaker scKO samples (*Figure 6*). Consequently, the enhanced accumulation of ECM in myomaker scKO MOV muscle may explain the increase in muscle weight despite the lack of myofiber hypertrophy (*Figure 3*). This non-functional gain in muscle size illustrates that in place of de novo and larger myofibers, there is an increased formation of connective tissue in muscles with non-fused MPs, suggesting a protective role for fusion in stabilizing muscle architecture associated with increased workload.

## Discussion

Skeletal muscle hypertrophy is a highly desirable adaptation of various therapeutic strategies to restore strength and function in muscle wasting conditions. Functional gains in adult muscle mass are derived from elevations in protein synthesis within the myofiber, and our findings highlight an additional source for muscle hypertrophy through the contributions of muscle stem cell fusion. Specifically, we use BrdU labeling and genetic lineage tracing approaches to evaluate the extent of fusion during overload-induced hypertrophy. While BrdU analysis revealed approximately 30% of fibers to undergo a fusion event, findings from the fluorescent reporter model demonstrate that the majority of myofibers fuse (mGFP$^+$), indicating the BrdU strategy is an underestimation of the fusion process during muscle hypertrophy. We also reveal that in response to increased muscle workload, myomaker is primarily upregulated in activated SCs, which is required for fusion with a myofiber. In the absence of SC-derived myomaker, the activation of pro-hypertrophic signaling pathways and protein synthesis, and increases in myofiber size and number are diminished. Thus, myomaker-mediated fusion is required for load-induced hypertrophy of skeletal muscle.

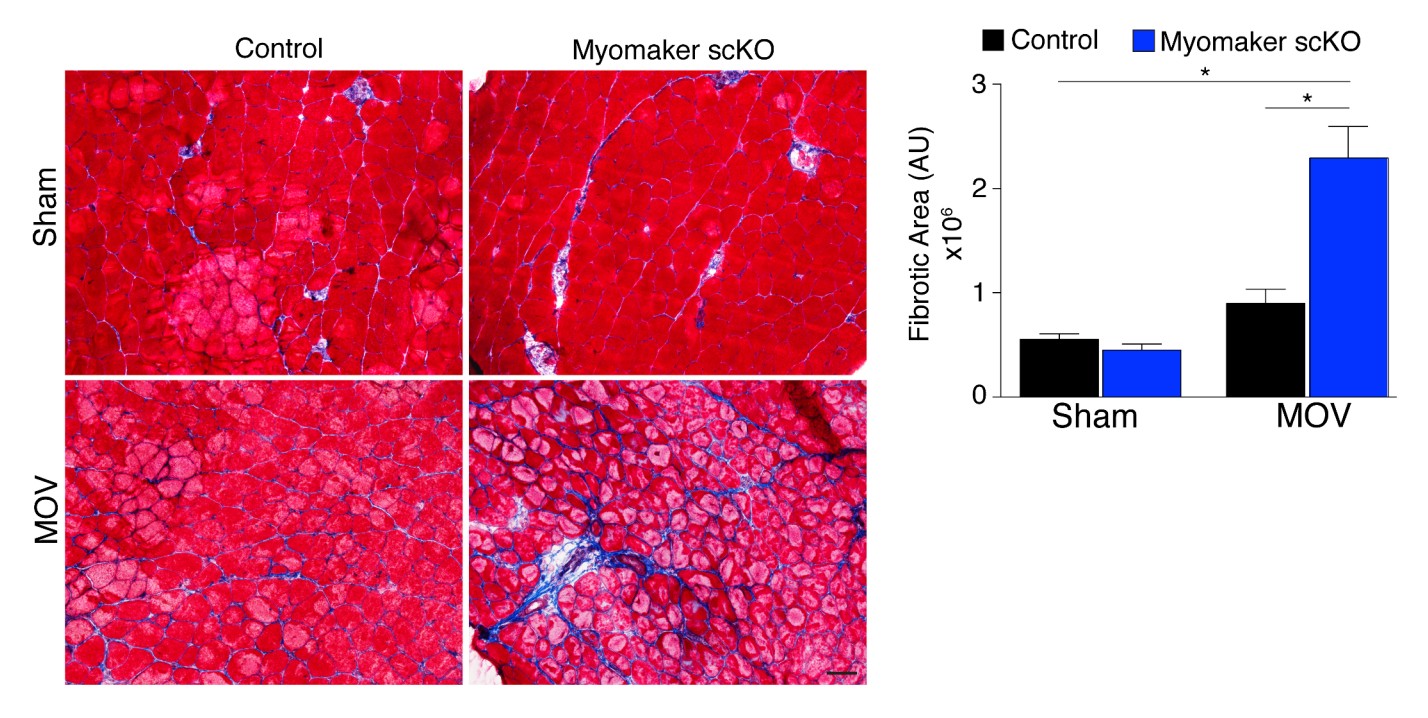

**Figure 6.** Loss of myomaker in satellite cells results in increased fibrosis during physiological hypertrophy. Trichrome stained sections from control and myomaker scKO mice demonstrate a significant increase in fibrosis in myomaker scKO mice after MOV. Quantification of the fibrotic area further demonstrates increased fibrosis in fusion-incompetent muscle. Values indicate mean (n = 4–12 mice) ± SEM, *p<0.05. Scale bar: 50 μm.

The requirement of muscle stem cells during adult skeletal muscle hypertrophy has long been a source of debate. Indeed, blunting of hypertrophy was observed after elimination of satellite cells through gamma irradiation (*Adams et al., 2002*) or through genetic manipulation of IL-6-dependent effects on SC/MP activity (*Guerci et al., 2012*; *Serrano et al., 2008*). However, two independent groups utilized DTA-mediated ablation of SCs and produced widely contrasting results. The initial SC DTA experiments indicated that SCs are not necessary for hypertrophy, where CSA was increased 10% in control mice with normal SC activity. (*Fry et al., 2014*; *McCarthy et al., 2011*). In contrast, a recent report also using ablation of SCs through DTA provides evidence that muscles fail to undergo overload-induced hypertrophy in the absence of SCs, and this study reported a 26% increase of CSA in control mice (*Egner et al., 2016*). While the interpretation of these two studies are vastly different, the magnitude of hypertrophy achieved, which is best assessed through CSA analysis and not gross muscle weights, is not similar and may account for the divergent outcomes. The magnitude of CSA increase in our study was 29% in control mice after MOV, which is more similar to the work by Egner et al. We also observed an approximate 10% CSA increase in myomaker scKO MOV samples compared to myomaker scKO sham. Although this increase was not significant it suggests that myofibers harbor a minor hypertrophic potential in the absence of fusion consistent with Fry et al. Through generation of fusion-defective SCs however, we provide independent evidence for the necessity of SCs for any major muscle hypertrophy.

While our findings demonstrate that fusion during muscle overload is inhibited by loss of myomaker in SCs, the exact contribution of fusion for hypertrophy remains to be elucidated. During compensatory hypertrophy, newly acquired myonuclei precede muscle hypertrophy (*Bruusgaard et al., 2010*). Thus, fusion may provide new nuclei to facilitate enhanced transcription of contractile proteins necessary for hypertrophy, as supported by our findings that myofibers are not able to activate protein synthesis in the absence of fusion. Addition of a new nucleus to the myofiber may also function to maintain a constant myonuclear domain (*Moss and Leblond, 1970*), however this concept has been challenged especially during postnatal muscle development (*White et al., 2010*). Alternatively, fusion may not only provide new nuclei but also other factors

necessary for hypertrophy, such as ribosomes for protein translation, or a bolus of signaling molecules to activate hypertrophic pathways similar to the activation of Akt by Wnt7a (*von Maltzahn et al., 2012*). These concepts clearly require more extensive work to reveal definitive mechanisms through which SC fusion drives adult skeletal muscle hypertrophy.

The Akt/mTOR pathway is controlled by insulin-like hypertrophy factor 1 (IGF-1) and myostatin, a member of the transforming growth factor-$\beta$ (TGF-$\beta$). IGF-1 directly activates the Akt pathway (*Rommel et al., 2001*) while myostatin is a negative regulator of muscle mass through SMAD-dependent inhibition of Akt (*Sartori et al., 2009*; *Trendelenburg et al., 2009*). The inability for muscle to initiate Akt/mTOR signaling without myomaker expression in myogenic progenitors was surprising since previous studies have shown that myofibers have the ability to intrinsically activate this canonical protein synthesis pathway (*Bodine et al., 2001*; *Lai et al., 2004*). Indeed, SC activation is not observed during muscle hypertrophy induced through a constitutively active Akt transgene (*Blaauw et al., 2009*), or inhibition of myostatin action by pharmacological blockade of its receptor ACVR2B (*Lee et al., 2012*). It is conceivable that during overload-induced hypertrophy, the increased tension placed on the myofiber requires SC-mediated activity for hypertrophy, whereas there is an absence of mechanical strain on the myofiber through treatment with IGF-1 or ACVR2B antibody. The ultimate goal of muscle hypertrophy is to increase contractile capability to deal with heightened physiological demands. Hypertrophy associated with loss of myostatin may not yield long-term improvements in muscle function suggesting that addition of myonuclei are necessary for functional hypertrophic gains (*Gundersen, 2016*). We did not assess muscle contractile function in our fusion-defective mice after MOV, however this will be important to fully elucidate the physiological relevance for MP fusion during muscle hypertrophy.

Our finding of increased fibrosis in muscles of myomaker scKO mice after overload further suggests a protective role for SC fusion on myofiber stability in response to mechanical strain, however the cellular circuitry driving fibrosis development remains unknown. DTA-mediated deletion of SCs also results in increased ECM components during overload-induced hypertrophy and aging, potentially through lack of interactions between SCs and fibroblasts (*Fry et al., 2014*; *Judson et al., 2013*; *Murphy et al., 2011*). Indeed, recent work suggests that the main function of MPs during overload-induced hypertrophy is to release miR-206-containing exosomes that dampen collagen production in fibroblasts, and that loss of MPs facilitates development of fibrosis (*Fry et al., 2017*). After overload of myomaker scKO mice, MPs are still present in muscle, suggesting the major reason for fibrosis can be attributed to an inability to maintain myofiber integrity due to loss of SC activity. It is formally possible that MPs lacking myomaker are not only fusion-defective but also lack an ability to communicate with fibroblasts through paracrine signaling. However, given that no evidence exists supporting this possibility, and that we observed significant fusion during MOV, our data are more consistent with a model where fibrosis occurs in myomaker scKO MOV muscles due to an inability to repair myofiber mechanical strain. This model of myofiber mechanical strain eliciting fibrosis is not unlike what occurs in other contexts, such as in cardiac muscle subjected to pressure overload where fibrosis functions to maintain tissue architecture (*Creemers and Pinto, 2011*; *Teekakirikul et al., 2012*).

One intriguing question to arise from this study pertains to the fate of fusion-incompetent MPs. Compared to control, we observed an increase in the number of LacZ$^+$ cells (MPs) in myomaker scKO muscle at day 7 after MOV. These data could indicate an accumulation of MPs because they have not fused. However, 10 days after MOV the number of LacZ$^+$ cells was comparable between control and myomaker scKO muscle, suggesting that accumulated LacZ$^+$ fusion-defective cells die between day 7 and day 10 of MOV or that myomaker is down-regulated and they become LacZ$^-$. Analysis of MPs using a genetic reporter (mGFP) found no dramatic difference in the percentage of mGFP$^+$ cells between control and myomaker scKO MOV samples. These inconsistencies highlight the current lack of knowledge regarding fusion-defective MPs. It is possible that the LacZ$^+$ cells represent a small fraction of MPs and not enough to shift the percentage of mGFP$^+$ MPs analyzed through flow cytometry. Also, since mGFP$^+$ MPs are expressed as a percentage of the total cell population, changes in non-MP populations could indirectly alter the mGFP$^+$ percentage. Nonetheless, the behavior and fate of fusion-incompetent MPs requires future investigation.

In conclusion, we have shown that myomaker is activated on SCs allowing fusion with a myofiber during muscle hypertrophy, thereby promoting hypertrophic signaling pathways and minimizing fibrosis associated with mechanical strain. These findings modify current theories for the role of

muscle stem cells during hypertrophy and may lead to new intervention strategies for muscle wasting diseases.

## Materials and methods

### Animals

The generation of myomaker mouse strains used in this study was described previously (*Millay et al., 2014*). Briefly, $Tmem8c^{LacZ/+}$ mice contain a LacZ cassette in intron 1 of the myomaker locus. $Tmem8c^{loxP/+}$ mice contain loxP sites that flank exon 2. $Tmem8c^{LacZ/loxP}$ mice were then bred with mice carrying the satellite cell-specific ($Pax7^{CreERT2}$) (*Lepper et al., 2009*) or myofiber-specific ($ACTA1^{CreERT2}$) (*Schuler et al., 2005*) Cre recombinase. To assess fusion, $Tmem8c^{+/+}$; $Pax7^{CreERT2/+}$ mice (control) and $Tmem8c^{LacZ/loxP}$; $Pax7^{CreERT2/+}$ mice (myomaker scKO) were crossed with $Rosa26^{mTmG}$ mice (Jackson Laboratory # 007676). Experimental mice were at least 3 months of age prior to any procedure performed. All animal procedures were approved by Cincinnati Children's Hospital Medical Center's Institutional Animal Care and Use Committee.

### Tamoxifen and BrdU

Tamoxifen (TAM; Sigma-Aldrich) was mixed in corn oil with 10% ethanol at 25 mg/ml. Prior to muscle overload surgery, 3-month old mice were administered five daily doses of TAM at 0.075 mg/g/day through intraperitoneal injections. Following surgery, mice were maintained on a TAM regimen (with the exception of mice used in *Figure 2C*) throughout MOV and administered 10 µl/g BrdU (Thermo Fisher Scientific) as indicated. No statistical differences in body weights were observed after tamoxifen treatment in either control or $myomaker^{scKO}$ mice.

### Synergist ablation

Muscle overload of the plantaris muscles was achieved through bilateral synergistic ablation of soleus and gastrocnemius muscles as described by others (*Dearth et al., 2013*). Briefly, the soleus and gastrocnemius muscles were exposed by making an incision on the posterior-lateral aspect of the lower limb. The distal and proximal tendons of the soleus were cut followed by cutting of the distal gastrocnemius tendon and excision of 75% lateral and medial gastrocnemius.

### Muscle collection and preparation

Plantaris muscles were dissected, blotted dry, and weighed. Hindlimbs were dissected and dissolved overnight in lysis buffer containing 0.4 mg/ml proteinase K at 55°C. Tibia length was assessed by a digital caliper, and the average of three measurements was recorded. For cryosections, muscles were embedded in 1% tragacanth/PBS (Sigma), and frozen in liquid nitrogen-cooled 2-methylbutane. Sections were then cut at 10 µm. For gene and protein analysis, mice were fasted for four hours, and muscles were flash-frozen in liquid nitrogen and stored at −80°C. For $Tmem8c^{LacZ/loxP}$; $Pax7^{CreERT2/+}$; $Rosa26^{mTmG}$ mice, plantaris muscles were fixed in 4% PFA/PBS for 5 hr at 4°C and sucrose protected overnight (30% sucrose/PBS) followed by embedding and cryosectioning.

### X-gal staining

Cryosections were stained with X-gal as described previously (*Millay et al., 2014*). After staining, the samples were rinsed in PBS and fixed in 4% PFA/PBS for at least 10 min. Tissue sections were co-stained with light eosin (0.7%), dehydrated, and mounted with Permount (Fisher). Samples were visualized on an inverted bright-field microscope (Olympus BX51), and the numbers of LacZ⁺ non-myofibers, LacZ⁺ myofibers, and all myofibers in a given field of view were manually counted. Data presented is representative of the average number of LacZ⁺ non-myofibers, and the percentage of LacZ-expressing myofibers, quantified from several non-overlapping fields of view.

### Muscle progenitor isolation and flow cytometry

We isolated skeletal muscle stem cells using techniques described previously (*Liu et al., 2015*). Briefly, plantaris muscles from either tamoxifen-treated $Tmem8c^{+/+}$; $Pax7^{CreERT2/+}$; $Rosa26^{mTmG}$ or $Tmem8c^{LacZ/loxP}$; $Pax7^{CreERT2/+}$; $Rosa26^{mTmG}$ mice were collected and minced in 10% horse serum (HS) (Gibco # 26050–070), and then incubated in 800 U/ml Collagenase Type 2 (Worthington # CLS-

2) solution at 37°C with gentle agitation for 1 hr. Following centrifugation, pellets were resuspended in 10% HS with 1000 U/ml Collagenase Type II and 4.8 U/ml Dispase (Roche # 4942078001), and incubated at 37°C with gentle agitation for another 30 min. Following this second round of incubation, samples were triturated with a 20-gauge needle, centrifuged and resuspended in 10% HS. Cell suspensions were subsequently filtered through a 40 μm nylon cell strainer (Corning # 352340), centrifuged, and resuspended in 2% Fetal Bovine Serum/PBS (Hyclone # SH30071). Flow cytometry analysis on cell suspensions was performed with a BD Biosciences LSR II Flow Cytometer configured with the 488 nm laser for GFP and the 561 nm laser for Tomato. Voltages were determined using cell suspensions from vehicle-treated $Tmem8c^{+/+}$; $Pax7^{CreERT2/+}$ and $Tmem8c^{+/+}$; $Pax7^{CreERT2/+}$; $Rosa26^{mTmG}$ mice. Analysis was performed using FACSDiva software.

## Histological analyses

Cryosections were fixed in 1% PFA/PBS; permeabilized with 0.2% Triton X-100/PBS or denatured with 2 M HCL in 0.5%Triton X-100/PBS and subsequently neutralized with TBS (pH 8.4) for BrdU labeling. After neutralization, sections were blocked with 1% BSA, 1% heat-inactivated goat serum, and 0.025% Tween20/PBS; and incubated with primary antibody overnight at 4°C. Following 1 hr of incubation with a secondary Alexa Fluor antibody (1:200) (Invitrogen), slides were mounted with VectaShield containing DAPI (Vector Laboratories). For assessment of hypertrophy, samples were stained with anti-laminin (1:100; Sigma-Aldrich # L9393), and representative non-overlapping images from all sections were captured at 20x on a Nikon A1R confocal system. The cross-sectional area (CSA) of myofibers was quantified through a binary thresholding algorithm developed in Imaris (Bitplane). Whole section images were also taken at 10x, and the total number of myofibers per muscle section was determined through an automated image segmentation program from ImageJ. To assay for fusion, samples were co-stained with anti-BrdU (1:20; Roche #11170376001) and anti-dystrophin (1:100; Abcam #ab15277). The following antibodies from Developmental Studies Hybridoma Bank were also used: Pax7 (1:50), myh7 (1:100; clone BA-D5), myh2 (1:100; clone SC-71), myh4 (1:10; clone BF-F3), and myh3 (1:10; clone F1.652). For Pax7 immunostaining, the above protocol was used in addition to antigen retrieval (1x Antigen Retrieval Citra Solution (Biogene # HK086-9K) and boiled for 30 min), followed by incubation with Pax 7antibodies overnight. The following day, 1:200 M.O.M. biotinylated anti-mouse IgG (Vecta Laboratories) was incubated for 30 min, washed, and then 1:200 Strepdavidin-Alexa Fluor 488 (Invitrogen) was incubated for 30 min. All samples were visualized at 40x with a Nikon A1R+ LUNV on a Ti-E Inverted Microscope unless otherwise noted. For mTom/mGFP analysis, sucrose-protected samples were visualized at 10x. The authenticity of mGFP signal was verified by establishing baseline levels for auto-fluorescence in vehicle-treated mice (GFP⁻). Masson's Trichrome staining was performed using standard protocols, and total fibrotic area within a section was quantified with a threshold intensity program from ImageJ.

## Single fiber isolation

To quantify overall mGFP⁺ expression, single fibers were isolated from plantaris muscles of sham and overloaded mice, as described previously (*Kanisicak et al., 2009*). Whole plantaris muscles were incubated in 0.2% Type I collagenase (Sigma-Aldrich, # C-0130) for 1 hr at 37°C. Muscles were transferred to DMEM (Hyclone) containing 10% horse serum and triturated to release individual fibers. Fibers were then fixed in 4% PFA/PBS for 30 min at 4°C, washed three times in PBS, permeabilized and mounted with Vectashield containing DAPI, and visualized at 10x on a Nikon A1R confocal system. Identical settings were utilized to capture mGFP and mTomato fluorescent images from all fibers and then processed in ImageJ. Fibers isolated from vehicle-treated mice were used to establish an impartial baseline for auto-fluorescence, and the threshold for GFP intensity was set at a constant value. The absolute GFP intensity for individual experimental fibers was then determined by tabulating the integrated density of saturated pixels above threshold levels. Similarly, tomato expression was used to quantify myofiber area by calculating the total pixelated area above background. GFP intensity was then normalized to myofiber area.

## In vivo assessment of protein synthesis

Changes in protein synthesis in whole muscle cross-sections were detected through utilization of antibiotic puromycin, a non-radioactive technique known as surface sensing of translation (SUnSET)

(*Goodman et al., 2011*; *Schmidt et al., 2009*). Following 4 hr of fasting, mice were administered 0.04 µmol puromycin/gram of body weight dissolved in PBS intraperitoneally for 30 min, and plantaris muscles subsequently collected for cryosections. Sections were immunostained with anti-puromycin (1:1000; EMD Millipore; clone 12D10), and anti-embryonic myosin (myh3). For the purpose of defining the myofiber borders we also stained the sections with anti-laminin. Representative images were captured at 20x, and a binary algorithm developed in Imaris was used to calculate the mean puromycin intensity within a laminin-stained myofiber. Samples were automatically distinguished as myh3$^+$ (de novo) or myh3$^-$ (existing) myofibers within the same image, and puromycin intensities for individual myofibers in each group were quantified separately.

## RNA analysis

Total RNA was extracted from plantaris muscle with Trizol (Invitrogen), and cDNA was synthesized using MultiScribe™ reverse transcriptase with random hexamer primers (Applied Biosystems). Gene expression was assessed using standard qPCR approaches with PowerUp™ SYBR Green Master Mix (Applied Biosystems). Analysis was performed on a 7900HT fast real-time PCR machine (Applied Biosystems) with the following myomaker SYBR primers: Forward, 5'-ATCGCTACCAAGAGGCGTT-3'; and Reverse, 5'-CACAGCACAGACAAACCAGG-3'. GAPDH primers (Forward, 5'-TGCGACTTCAACAGCAACTC-3'; and Reverse, 5'-GCCTCTCTTGCTCAGTGTCC-3') were used as internal controls.

## Western blot

A myomaker custom antibody was generated through YenZym Antibodies LLC. Rabbits were immunized with amino acids #137–152 of mouse myomaker (MKEKKGLYPDKSIYTQ) after conjugation to KLH. We used antigen-specific affinity purified products for western analysis. Plantaris muscles were homogenized with a bead homogenizer (TissueLyser II; Qiagen) in a lysis buffer [10 mM Tris (pH 7.4), 1 mM EDTA, 1 mM dithiothreitol, 0.5% Triton X-100, 2.1 mg/ml NaF] containing protease and phosphatase inhibitor cocktails (5 µl/ml; Sigma-Aldrich), and following centrifugation, the amount of protein in supernatants was determined through Bradford protein assay. Muscle homogenates were heated at 65°C for 30 min, separated on 10% (30 µg of protein for detection of signaling proteins along the Akt/mTOR pathway) or 12% (100 µg of protein for myomaker detection) SDS-page gels, and transferred to PVDF-FL membranes (Immobilon). Membranes were subsequently blocked in 5% dry milk or 5% BSA in Tris-buffered saline (TBS)-Tween, and incubated overnight at 4°C with an antibody against myomaker (1:200), phosphorylated Akt (Ser473) (1:750; Cell Signaling # 9271), total Akt (1:750; Cell Signaling # 9272), phosphorylated p70s6k (Thr389) (1:500; Santa Cruz # SC-11759), or total p70s6k (1:750; Cell Signaling # 2708). GAPDH (1:5000; Millipore) served as a control for sample loading. Membranes were then washed and incubated with an IRDye 800CW anti-rabbit secondary antibody (1:5000; LI-COR Biosciences). The relative abundance of phosphorylated and total protein levels of Akt and p70s6k was quantified using the Odyssey infrared detection system (LI-COR Biosciences).

## Statistical analysis

Quantitative data sets are presented as means ± SEM, and analyzed with an unpaired t-test (*Figures 1D* and *2A*), a one-way analysis of variance (ANOVA) (*Figures 1A, C* and *2C*), or a two-way ANOVA (all other data sets) using GraphPad Prism 6 software. For assessments of myofiber CSA, total fiber number, BrdU incorporation, single fiber fluorescent intensity, and levels of protein synthesis in myofibers both image captures and data analyses were performed in a blinded fashion. Statistical significance was set at a p value <0.05.

## Acknowledgements

We acknowledge the following entities within Cincinnati Children's Hospital Medical Center: Onur Kanisicak and Vikram Prasad of the Molkentin laboratory for technical advice, the Confocal Imaging Core for microscopy assistance, the Heart Institute Biorepository for Masson's Trichrome staining, and the Research Flow Cytometry Core for FACs assistance. This work was supported by grants to DPM from the Cincinnati Children's Hospital Research Foundation, National Institutes of Health (R01AR068286), Muscular Dystrophy Association, and Pew Charitable Trusts.

# Additional information

## Funding

| Funder | Grant reference number | Author |
|---|---|---|
| Muscular Dystrophy Association | | Douglas P Millay |
| Pew Charitable Trusts | | Douglas P Millay |
| National Institute of Arthritis and Musculoskeletal and Skin Diseases | R01AR068286 | Douglas P Millay |

The funders had no role in study design, data collection and interpretation, or the decision to submit the work for publication.

## Author contributions

QG, Conceptualization, Formal analysis, Investigation, Writing—original draft, Writing—review and editing; DPM, Conceptualization, Data curation, Formal analysis, Supervision, Funding acquisition, Investigation, Methodology, Writing—original draft, Project administration, Writing—review and editing

## Author ORCIDs

Douglas P Millay, http://orcid.org/0000-0001-5188-0720

## Ethics

Animal experimentation: This study was performed in strict accordance with the recommendations in the Guide for the Care and Use of Laboratory Animals of the National Institutes of Health. All of the animals were handled according to approved institutional animal care and use committee (IACUC) protocols (#2014-0051) of the Cincinnati Children's Hospital Medical Center. All surgery was performed under isoflurane anesthesia, and every effort was made to minimize suffering.

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
