## [Decision Letter]

Thank you for submitting your article "Requirement of myomaker-mediated stem cell fusion for skeletal muscle hypertrophy" for consideration by *eLife*. Your article has been favorably evaluated by Sean Morrison (Senior Editor) and three reviewers, one of whom, Amy J Wagers (Reviewer #1), is a member of our Board of Reviewing Editors.

The reviewers have discussed the reviews with one another and the Reviewing Editor has drafted this decision to help you prepare a revised submission.

Summary:

Pax7^+^ Satellite cells (SCs) are resident muscle stem cells that are indispensable for adult skeletal muscle regeneration. Whether SCs are required for muscle fiber hypertrophy remains controversial. A previous study using Pax7-DTA alleles (*Pax7*^CreER/+^; *Rosa26*^DTA/+^) to deplete SCs showed no impact on subsequent fiber hypertrophy upon mechanical overload (MOV) despite a diminished increase of myonuclei (McCarthy et al. 2011). However, results from this study could have been confounded by incomplete SC ablation and/or by indirect effects of dying SCs.

Goh and Millay address this ongoing controversy from a new angle, by generating fusion-incompetent muscle progenitors via manipulation of expression of myomaker, a recently identified transmembrane protein necessary for myoblast fusion. The authors use state-of-the-art genetic systems, including conditional myomaker alleles to achieve SC-specific deletion of myomaker (myomaker^scKO^) in adult mice. They show that with these animals, in which SCs and their progeny are unable to fuse, overload-induced hypertrophy is blunted, indicating a role for SC fusion in mounting an optimal hypertrophic response. They substantiate this conclusion using additional BrdU molecular tracking and a color-switching genetic lineage tracing strategies, which further demonstrate the contribution and requirement of myomaker-mediated stem cell fusion for skeletal muscle hypertrophy. Finally, the authors provide data indicating a reduction of Akt signaling, a pathway known for eliciting muscle hypertrophy, in myomaker scKO MOV muscles following fusion-blockade (although this aspect of the work remains rather preliminary, see below).

Overall, the work has a number of strengths. Most importantly, the novel approach adopted by the authors to disable SC fusion without directly inducing cytotoxicity offers a fresh perspective to study the requirement of SCs for muscle hypertrophy. The experiments are well designed and conducted, and the arguments are reasonably supported. The genetic tools developed by the authors could be useful beyond the scope of this manuscript to the skeletal muscle field for studying the contribution of muscle stem cells fusion in other settings. There are however a number of issues related to the strength with which the authors' data support the conclusions drawn and the presentation of these data that need to be addressed before publication.

Essential revisions:

1) Ideally, the impact of fusion blockade by ablation of myomaker in muscle progenitor cells would be further supported by evaluation of muscle contractile function. This is important in order to establish the essential contribution of myomaker-mediated stem cell fusion during physiological muscle hypertrophy. Both whole-muscle and single-fiber function assays have been reported previously for MOV systems (Fry et al. 2014 FASEB journal), and inclusion of one of these types of analyses would greatly strengthen the paper. However, the reviewers recognize that these studies may prove difficult to complete in a timely fashion, and so, if the authors are unable to include them in a revised manuscript, they should instead add a comment to the paper's discussion indicating that such physiology studies will be important to evaluate in future studies.

2) In Figure 1, and E, how do the authors recognize mononuclear cells and fully differentiated myocytes? How do they distinguish the location of blue color in day 3, 7, 10 and 14 in Figure 1? Lots of blue color appears inside cells, not on the membrane. Does this mean that myomaker is not a membrane protein in myotubes? These are critical points and should be explained and documented by clear experiments.

3) The authors rely extensively on reporter gene expression and mRNA () to track Myomaker protein; however, as commercial antibodies are now available, they should confirm these data by direct staining for myomaker protein in SCs and myofibers.

4) Analysis of co-localization of Pax7 and lacz^+^ in MOV model using the existing subsequent muscle sections is needed to evaluate if satellite cell homing/localization is affected in the hypertrophy model.

5) There are multiple muscle fiber types present in the plantaris muscle, including types I, IIa, IIb, IIa/b and IIx. Different fiber types are known to have distinct CSAs and a shift of relative fiber type frequency among different experimental and control groups could potentially confound the quantitative analysis for CSA and fiber numbers (Figure 3) especially when only a subset of the section is analyzed. Particularly, Figure 3 did not show appreciable difference between the fiber CSA of *myomaker*^scKO^ MOV and Sham controls but a clear shift towards larger myofiber sizes of *myomaker*^scKO^ MOV is observed in 3E. An analysis of relative fiber type frequency and the CSA of each fiber type is necessary for clarifying and validating the quantitative claims and achievable by re-staining the current section slides.

6) Figure 2 shows an increase at day 7 and then dramatic decrease of lacz^+^ mononuclear cells by day 10 after MOV. The authors effectively show that the accumulated lacz^+^ cells do not fuse into fibers, but they have not explained the nearly 10-fold reduction that occurs in just 3 days. Is there evidence of lacz^+^ (or lacz^-^) muscle progenitor cell apoptosis during this interval, or are the cells simply shutting down expression of myomaker? Clarification of this point is important for interpreting the results, and could be obtained by (1) an analysis of lacz^+^ MP apoptosis rates at multiple time points after MOV (days 3, 7, 10 and 14) and (2) analysis of MP cell number and apoptosis rates at additional time points after MOV using the mTmG model, in which cells are permanently marked with GFP, so should be trackable regardless of whether they downregulate myomaker. Relatedly, an inherent assumption in the authors' discussion of contrasting results in the DTA ablation model versus myomaker deletion model is that myomaker deletion does not affect the frequency of muscle stem cells (satellite cells) in MOV muscles; however, this is not demonstrated directly. The authors should quantify satellite cell numbers in scKO versus control MOV muscles to directly evaluate effects of myomaker loss in the MOV model on maintenance of muscle regenerative cells.

7) The authors appear to use the same results in different ways, generating more figure panels (e.g. Figure 1, Figure 2, Figure 3, Figure 4, Figure 5 and Figure 6). While this approach is used occasionally in the literature to help the reader's understanding, in this paper, it's overused. Data presentation should be revised so that one source of results is used for one figure panel.

8) In Figure 3, fusion of MP to myofiber significantly increased fiber number whereas myomaker KO in SC decreased the levels to that of the control. If two cells fuse, shouldn't the produce be one cell in a larger volume, as shown by a dramatic increase in muscle mass (Figure 3)? Does this mean that myomaker is required for the increase in fiber number? The authors do not explain this important observation, and rather described it as an inability of MP to fuse. They should explain this finding about the role of myomaker in this process more clearly. This result shows that myomaker is required not only for the hypertrophy but also for the increase in fiber number.

9) Because the results in Figure 4 show that 30% of myofibers are engaged in fusion, the ~2.5 fold increase in muscle size in Figure 3 seems to be a quite surprising. The increase in Fiber CSA seems to be consistent with Figure 4. Does this mean myofibers become much denser? The authors should address this gap between the marked increase (2.5-fold) in muscle mass and the relatively small fraction (30%) of myotubes involved infusion.

10) Figure 5 seems to be very superficial, although the control of hypertrophy by myomaker is the main point of this manuscript. This pathway, AKT/mTOR, exists not only in the progenitor but also in myofibers. Thus, it is not new that the signaling pathway (i.e. p-AKT, p-S6K) is decreased by compromised hypertrophy. This should be discussed in the manuscript, along with the need to uncover the mechanistic basis for this result.

[Editors' note: further revisions were requested prior to acceptance, as described below.]

Thank you for resubmitting your work entitled "Requirement of myomaker-mediated stem cell fusion for skeletal muscle hypertrophy" for further consideration at *eLife*. Your revised article has been favorably evaluated by Sean Morrison (Senior Editor), and Amy Wagers (Reviewing Editor).

We appreciate your responsiveness to the prior review, and agree that the manuscript has been substantially improved. However, there are some remaining issues that need to be addressed before acceptance, as outlined below:

1) The authors’ conclusion that satellite cell frequency is not impacted in the scMyomaker mice under MOV (Figure 2—figure supplement 2) is not convincing because (1) the representative FACS plots show a very apparent loss of GFP^+^ cells in the scMyomaker group, even if the quantification indicates that this is not significant for n=3 mice, and a quick power analysis of this experiment, assuming that error bars represent SEM, suggests that the study may be underpowered to detect a difference, and more animals should be analyzed, (2) it appears from the Methods that the authors have not included live/dead discrimination in the FACS gating strategy, which will increase "noise" in their analyses by including variable amounts of debris and dead cells, and (3) as the authors appropriately point out in the Discussion, their analysis of% GFP^+^ cells is influenced not only by possible changes in the satellite cell subset, but also by changes in the numbers of non-myogenic (inflammatory, fibrotic, etc.) cells, which could be influenced indirectly by fusion-deficits caused by myomaker deletion. With all of these caveats, it seems inappropriate to draw a strong conclusion from these data. Furthermore, the Pax7 immunostaining data provided by the authors in support of their conclusion shows only single fields, and is not quantified. The authors should address this concern by either improving the robustness of their FACS analysis or providing quantification of the Pax7 immunostaining assays (or both).

2) The authors need to specify in every figure legend (including supplemental figures) whether error bars represent SD or SEM.

3) Regarding the statement in the Results that "30% of fibers underwent fusion", the authors should include comment that this is likely an underestimate (and explain why) in the Results section (subsection “Fusion during MOV is blocked through deletion of myomaker in SCs”) when these data are first discussed. They do this well in the Discussion, but it should be noted at the time the data are presented as well.

---

## [Author Response]

*Essential revisions:*

*1) Ideally, the impact of fusion blockade by ablation of myomaker in muscle progenitor cells would be further supported by evaluation of muscle contractile function. This is important in order to establish the essential contribution of myomaker-mediated stem cell fusion during physiological muscle hypertrophy. Both whole-muscle and single-fiber function assays have been reported previously for MOV systems (Fry et al. 2014 FASEB journal), and inclusion of one of these types of analyses would greatly strengthen the paper. However, the reviewers recognize that these studies may prove difficult to complete in a timely fashion, and so, if the authors are unable to include them in a revised manuscript, they should instead add a comment to the paper's discussion indicating that such physiology studies will be important to evaluate in future studies.*

We completely agree that whole-muscle and/or single-fiber assays for muscle contractile function would be valuable in establishing greater physiological relevance for myomaker-mediated stem cell fusion during overload-induced muscle hypertrophy. However, as mentioned by the reviewers we are unable to complete such experiments in a timely manner but have included a comment in our Discussion on the need for assessing contractile function in follow-up studies (fourth paragraph).

*2) In Figure 1, and E, how do the authors recognize mononuclear cells and fully differentiated myocytes? How do they distinguish the location of blue color in day 3, 7, 10 and 14 in Figure 1? Lots of blue color appears inside cells, not on the membrane. Does this mean that myomaker is not a membrane protein in myotubes? These are critical points and should be explained and documented by clear experiments.*

We agree with the reviewers that more detailed investigations are justified to distinguish the various LacZ^+^ cells. We have performed more experiments to address these concerns and also clarified the nature of the *Tmem8c*^LacZ/+^ mouse, which contains a *LacZ* cassette in intron 1 of the myomaker locus. Directly upstream of *LacZ* is a strong splice acceptor site that enables splicing of exon 1 of myomaker and *LacZ*, and an IRES sequence, which allows independent translation of *LacZ*. This allele does not result in a myomaker-LacZ fusion protein, and therefore does not function as a readout for myomaker localization. We only use this allele as a surrogate for myomaker transcription. We have explained these issues in great detail in the Results section (subsection “Dynamic expression of myomaker during muscle overload coincides with fusion”, first paragraph).

The reviewers make an excellent point that our initial description of LacZ^+^ cells did not accurately distinguish mononuclear cells and fully differentiated myocytes. Specifically, we observe various types of LacZ^+^ structures after MOV and in this revised manuscript attempted to more clearly articulate the main point of this figure – that after MOV myomaker expression initially occurs in MPs and then at later stages in myofibers. To deal with this issue we sought to delineate the myogenic state of these LacZ^+^ cells through determining whether these cells were myh3^+^ (embryonic myosin), a known marker for differentiated myocytes. Through staining of day 7 MOV serial sections with either x-gal or myh3 we were able to classify the LacZ^+^ structures. These results are displayed in Figure 1—figure supplement 1 and explained in the Results section (subsection “Dynamic expression of myomaker during muscle overload coincides with fusion”, first paragraph). Based on these observations, we have identified the following populations of x-gal^+^ cells:a) Myofibers that exhibit punctate LacZ and are myh3-. We interpret these cells to be existing myofibers that have fused with a LacZ^+^ MP, and the reason for the punctate LacZ is that after fusion LacZ diffuses throughout the large myofiber.

b) Myofibers that are both LacZ^+^ and myh3^+^, which represent de novo fibers because they express embryonic myosin. These myofibers exhibit stronger LacZ expression, compared to LacZ^+^ myh3^-^ myofibers, because they are formed from the fusion of multiple LacZ^+^ MPs and therefore generate a structure with concentrated LacZ.c) Small LacZ^+^ cells that were either myh3^+^ or myh3^-^. The small LacZ^+^ myh3^+^ cells are differentiated myocytes but they were not large enough to be classified as myofibers as in (b). We interpret the small LacZ^+^ myh3^-^ cells to be MPs given that myomaker is muscle-specific.

For quantitative purposes, we classified the LacZ^+^ cells in (a) and (b) above as ‘myofibers’, and the small LacZ^+^ cells described in (c) as ‘non-myofibers’. We acknowledge there could be many more questions about these data relating to the dynamics of myomaker expression in myogenic lineage progression, however this is not the central point of our manuscript and we think the revised data more effectively substantiates the overall meaning of the figure.

*3) The authors rely extensively on reporter gene expression and mRNA () to track Myomaker protein; however, as commercial antibodies are now available, they should confirm these data by direct staining for myomaker protein in SCs and myofibers.*

We indeed have solely analyzed myomaker expression using qPCR and our LacZ reporter mouse. The reviewers are also correct in that there are commercial antibodies available, and we have also generated two custom antibodies that recognize independent myomaker epitopes. We have tested multiple commercial antibodies as well as our custom antibodies, and these work for western blot and immunostaining on cultured myoblasts. However, the functionality of the antibodies to detect myomaker in vivo on sections is not clear as we observe staining in myomakerscKO samples that we have definitively shown are negative for myomaker mRNA. Moreover, we also observe widespread staining in control sham samples, which is problematic since we have previously demonstrated that unperturbed muscle is myomaker negative through qPCR and LacZ analysis. Therefore, we do not feel comfortable using these reagents to definitively track myomaker protein in SCs and myofibers. While we are attempting to optimize the antibodies for immunostaining, there is no guarantee of success and we think our genetic data in myomakerscKO and myomakermKO mice are sufficient for us to conclude that myomaker is not transcribed from a myonucleus of an existing myofiber during MOV.

*4) Analysis of co-localization of Pax7 and* lacz^+^*in MOV model using the existing subsequent muscle sections is needed to evaluate if satellite cell homing/localization is affected in the hypertrophy model.*

The reviewers raise an important question here. We initially did not perform immunostaining for Pax7 in control or myomakerscKO muscle after MOV. In our revised manuscript, we show that the number and localization of Pax7 cells are not grossly altered in myomakerscKO muscle, compared to control (Figure 2—figure supplement 2). There could be a small reduction in the number of Pax7 cells in myomakerscKO samples, this is unlikely to be a major factor contributing to the observed hypertrophy defect. Furthermore, our data that loss of myomaker in SCs inhibits fusion during MOV suggests compromised fusion is the central mechanism by which myomakerscKO muscle fails to hypertrophy.

*5) There are multiple muscle fiber types present in the plantaris muscle, including types I, IIa, IIb, IIa/b and IIx. Different fiber types are known to have distinct CSAs and a shift of relative fiber type frequency among different experimental and control groups could potentially confound the quantitative analysis for CSA and fiber numbers (Figure 3) especially when only a subset of the section is analyzed. Particularly, Figure 3 did not show appreciable difference between the fiber CSA of myomakerscKO MOV and Sham controls but a clear shift towards larger myofiber sizes of myomakerscKO MOV is observed in 3E. An analysis of relative fiber type frequency and the CSA of each fiber type is necessary for clarifying and validating the quantitative claims and achievable by re-staining the current section slides.*

We thank the reviewers for these remarks, which compelled us to more closely analyze our CSA data shown in revised Figure 3. The average myofiber CSA in myomakerscKO MOV compared to myomakerscKO sham actually did increase approximately 10% (Figure 3), however this was not statistically significant. This is potentially the reason of a shift to larger myofibers, although none of these data points are statistically significant. Overall, these data could suggest that fusion-defective muscle has a minor capacity for hypertrophy and we have now included a paragraph in the Discussion (second paragraph) dealing with this possibility.

Additionally, based on the reviewers suggestions we analyzed the CSA of various fiber types. We stained our muscle sections for type I, type IIa, type IIb, and type IIx muscle fibers, and quantified the CSAs and relative frequency for each fiber type (Figure 3—figure supplement 1). Our CSA analysis for the glycolytic fiber types (types IIa, IIb, and IIx) correspond to our main findings of significantly increased myofiber growth in overloaded control muscles, concurrent with impaired myofiber size exhibited by fusion-incompetent muscles across these fiber types. Furthermore, as we observe no more than 2% of the oxidative type 1 fibers in both control and myomaker^scKO^ muscles after overload, we conclude that the majority of myofibers in the plantaris muscle fail to hypertrophy effectively in the absence of satellite cell fusion, regardless of specific fiber types. Despite marked differences in fiber type CSA, relative frequency distribution analyses did not reveal a significant shift towards any particular fiber type after 2 weeks of muscle overload.

*6) Figure 2 shows an increase at day 7 and then dramatic decrease of* lacz^+^*mononuclear cells by day 10 after MOV. The authors effectively show that the accumulated* lacz^+^*cells do not fuse into fibers, but they have not explained the nearly 10-fold reduction that occurs in just 3 days. Is there evidence of* lacz^+^*(or lacz^-^) muscle progenitor cell apoptosis during this interval, or are the cells simply shutting down expression of myomaker? Clarification of this point is important for interpreting the results, and could be obtained by (1) an analysis of* lacz^+^*MP apoptosis rates at multiple time points after MOV (days 3, 7, 10 and 14) and (2) analysis of MP cell number and apoptosis rates at additional time points after MOV using the mTmG model, in which cells are permanently marked with GFP, so should be trackable regardless of whether they downregulate myomaker. Relatedly, an inherent assumption in the authors' discussion of contrasting results in the DTA ablation model versus myomaker deletion model is that myomaker deletion does not affect the frequency of muscle stem cells (satellite cells) in MOV muscles; however, this is not demonstrated directly. The authors should quantify satellite cell numbers in scKO versus control MOV muscles to directly evaluate effects of myomaker loss in the MOV model on maintenance of muscle regenerative cells.*

We thank the reviewers for these valuable comments. We have assessed the presence of muscle progenitors in control and myomakerscKOmuscle after MOV using two independent approaches. First, we used FACs to quantify the percentage of mGFP^+^ MPs, as suggested by the reviewers, and found no significant differences at day 7 or day 10 after MOV between control and myomakerscKOsamples (Figure 2—figure supplement 2). Second, we performed Pax7 immunostaining and also did not observe a major difference in Pax7^+^ cells during MOV (Figure 2—figure supplement 2). These data demonstrate that MPs are present in similar quantities, excluding this as a possibility that could underlie the hypertrophy defect.

The reviewers also question the mechanisms by which LacZ^+^ cells are reduced 10 days after MOV, compared to 7 days. We have included a new paragraph in the Discussion (sixth paragraph) dealing with these points.

The reviewers specifically wondered if myomakerscKO LacZ^+^ cells undergo apoptosis between day 7 and 10 of MOV. We assessed this possibility through TUNEL staining as well as a FACs-based cell death approach. We have provided these data for the reviewers (below) but would prefer not to include in the manuscript. The data do not indicate that cell death is the mechanism for the reduction of LacZ^+^ cells, although there is a non-significant increase in% Dead GFP^+^ myomakerscKOMPs. We have also previously shown an increase in TUNEL^+^ cells in developing limbs of myomaker KO mice, which combined with the modest increase in% Dead GFP^+^ cells below could suggest that fusion-defective cells are more prone to cell death. Because there are multiple modes of cell death that should be analyzed through independent strategies, our work examining this process is too preliminary to include, even as a supplemental figure. We think the fate of fusion-defective MPs is an extremely important question that will require more extensive analysis prior to a definitive conclusion. Another reason for our desire to omit the data is that the main point of the manuscript is that SC-mediated fusion is required for overload-induced hypertrophy, not the fate of fusion-defective MPs.

Author response image 1.Analysis of cell death after MOV.(**A**) Representative images of TUNEL^+^nuclei in plantaris sections demonstrates similar rates of apoptosis in control and myomaker^scKO^ muscles following MOV. (**B**) Flow cytometry to distinguish live and dead mGFP^+^ progenitors using a dye that binds to compromised membranes (LIVE/DEAD Fixable Stain, Invitrogen) in tamoxifen-treated *Pax7*^CreERT2/+^; *Rosa*^mTmG^ (control) and *myomaker*^LacZ/loxP^; *Pax7*^CreERT2/+^; *Rosa*^mTmG^ mice (myomaker^scKO^) highlights a small population of dead GFP^+^ cells 7 and 10d after MOV. Quantitative analysis reveals a modest (although non-significant) level of cell death in MPs between control and myomaker^scKO^ muscles at 7 and 10d of MOV (n = 3 mice/group). Scale bar: 10 μm.**DOI:**
http://dx.doi.org/10.7554/eLife.20007.014

*7) The authors appear to use the same results in different ways, generating more figure panels (e.g. Figure 1, Figure 2, Figure 3, Figure 4, Figure 5 and Figure 6). While this approach is used occasionally in the literature to help the reader's understanding, in this paper, it's overused. Data presentation should be revised so that one source of results is used for one figure panel.*

We note the overuse of figure panels in our manuscript, and have revised our data presentation to reflect one source of results for each figure panel. In addition, corresponding figure legends have been revised accordingly to reflect these changes.

*8) In Figure 3, fusion of MP to myofiber significantly increased fiber number whereas myomaker KO in SC decreased the levels to that of the control. If two cells fuse, shouldn't the produce be one cell in a larger volume, as shown by a dramatic increase in muscle mass (Figure 3)? Does this mean that myomaker is required for the increase in fiber number? The authors do not explain this important observation, and rather described it as an inability of MP to fuse. They should explain this finding about the role of myomaker in this process more clearly. This result shows that myomaker is required not only for the hypertrophy but also for the increase in fiber number.*

The reviewers are accurate and we have similarly interpreted our data, that myomaker is required for an increase in fiber number. We have highlighted this point in our manuscript (subsection “Satellite cell-derived myomaker is required for overload-induced muscle hypertrophy”, last paragraph and Discussion, first paragraph). Our current stance on physiological muscle growth in mice is that the hypertrophic response to overload comprises both myofiber growth and the formation of de novo myofibers (hyperplasia), as demonstrated in Figure 3 and Figure 3, respectively. Both these processes are governed by fusion; myofiber growth through fusion of satellite cells to an existing fiber, and hyperplasia through fusion of satellite cells to each other to generate de novo fibers. The increase in fiber number after MOV has been proposed to result from de novo myofiber formation or splitting of existing myofibers. However, to our knowledge, the evidence for myofiber splitting is weak suggesting that increased fiber number is attributed to de novo myofiber formation.

*9) Because the results in Figure 4 show that 30% of myofibers are engaged in fusion, the ~2.5 fold increase in muscle size in Figure 3 seems to be a quite surprising. The increase in Fiber CSA seems to be consistent with Figure 4. Does this mean myofibers become much denser? The authors should address this gap between the marked increase (2.5-fold) in muscle mass and the relatively small fraction (30%) of myotubes involved infusion.*

In the original manuscript, Figure 4 (Figure 4 in the revised version) shows that 30% of myofibers have undergone fusion, which was assessed through BrdU labeling. Given that this is one level of a particular muscle, if a BrdU myonucleus was above or below this plane then that fiber would not be counted as a positive fusion event. Therefore, this BrdU analysis underestimates the extent of fusion that occurs during MOV. Indeed, our lineage tracing analysis (Figure 4) shows that the majority of fibers fuse with MPs in response to MOV. We mention these details in the Discussion (first paragraph).

The reviewers bring up an excellent point regarding the discrepancy between muscle weights (Figure 3) and myofiber CSA (Figure 3). The main reason for this apparent inconsistency is that muscle weights are not the ideal readout for hypertrophy. We originally reported our muscle weights and CSA results as indicators of hypertrophy but have more clearly defined these analyses in the revised manuscript (Results, subsection “Satellite cell-derived myomaker is required for overload-induced muscle hypertrophy”, first paragraph). Muscle weights directly measure all components of the tissue, including myofibers, ECM, and immune cell infiltration and do not directly evaluate myofiber size. Moreover, since we remove two-thirds of the gastrocnemius during MOV, it is difficult to reliably dissect the plantaris away from the remaining gastrocnemius. This phenomenon could lead to the disparity between muscle weights and CSA, and reinforces that muscle weights should not be used as a proxy for hypertrophy. In contrast, the most direct assessment of hypertrophy is CSA, which we have emphasized in the revised manuscript.

*10) Figure 5 seems to be very superficial, although the control of hypertrophy by myomaker is the main point of this manuscript. This pathway, AKT/mTOR, exists not only in the progenitor but also in myofibers. Thus, it is not new that the signaling pathway (i.e. p-AKT, p-S6K) is decreased by compromised hypertrophy. This should be discussed in the manuscript, along with the need to uncover the mechanistic basis for this result.*

We agree that findings of associated reductions in Akt/mTOR signaling in fusion-incompetent muscles that demonstrate impaired hypertrophy following mechanical overload would be strengthened by more experiments. Hence, we have included additional experiments to further establish that myomaker-mediated stem cell fusion may augment physiological muscle growth. Specifically, we utilized the antibiotic puromycin to assess muscle protein synthesis, a downstream effector of Akt/mTOR signaling that directly leads to hypertrophy. Here, we report that levels of protein synthesis are elevated in control muscles at both 7 days and 14 days after overload. In contrast, the ability to synthesize proteins is compromised in overloaded myomaker^scKO^ fusion-incompetent muscles (Figure 5). These new data suggests a critical role for myonuclear accretion in generating requisite amounts of proteins to meet the increased physiological demands of mechanical overload. Per the reviewers’ request, we have also included a line in our manuscript on the need to further uncover the mechanisms by which satellite cell fusion drives muscle hypertrophy (Discussion, third paragraph).

[Editors' note: further revisions were requested prior to acceptance, as described below.]

*We appreciate your responsiveness to the prior review, and agree that the manuscript has been substantially improved. However, there are some remaining issues that need to be addressed before acceptance, as outlined below:*

*1) The author's conclusion that satellite cell frequency is not impacted in the scMyomaker mice under MOV (Figure 2—figure supplement 2) is not convincing because (1) the representative FACS plots show a very apparent loss of* GFP^+^*cells in the scMyomaker group, even if the quantification indicates that this is not significant for n=3 mice, and a quick power analysis of this experiment, assuming that error bars represent SEM, suggests that the study may be underpowered to detect a difference, and more animals should be analyzed, (2) it appears from the Methods that the authors have not included live/dead discrimination in the FACS gating strategy, which will increase "noise" in their analyses by including variable amounts of debris and dead cells, and (3) as the authors appropriately point out in the Discussion, their analysis of%* GFP^+^*cells is influenced not only by possible changes in the satellite cell subset, but also by changes in the numbers of non-myogenic (inflammatory, fibrotic, etc.) cells, which could be influenced indirectly by fusion-deficits caused by myomaker deletion. With all of these caveats, it seems inappropriate to draw a strong conclusion from these data. Furthermore, the Pax7 immunostaining data provided by the authors in support of their conclusion shows only single fields, and is not quantified. The authors should address this concern by either improving the robustness of their FACS analysis or providing quantification of the Pax7 immunostaining assays (or both).*

We agree with the reviewers that there are caveats associated with our FACS analysis, which is why we originally used a secondary method (Pax7 immunostaining) to confirm these data. In this revised version, we have improved our Pax7 immunostaining through use of a biotin-strepavidin detection system (Methods) and quantified the number of Pax7^+^ cells (Figure 2—figure supplement 2). Specifically, we express the number of Pax7^+^ cells per section or normalize to fiber number. That we do not see a significant difference in SC number when normalized to fiber number suggests that the muscle stem cells are not dramatically reduced in myomaker scKO muscle. We agree with the reviewers that it may be inappropriate to draw stronger conclusions from this data, other than there may be subtle alterations in SC dynamics but the myomaker scKO model does not significantly alter SC frequency. We have scaled back our interpretations of these data within the Results section (subsection “Myomaker is activated primarily in muscle progenitors”, first paragraph).

*2) The authors need to specify in every figure legend (including supplemental figures) whether error bars represent SD or SEM.*

We have now specified whether error bars represent SD or SEM within each figure.

*3) Regarding the statement in the Results that "30% of fibers underwent fusion", the authors should include comment that this is likely an underestimate (and explain why) in the Results section (subsection “Fusion during MOV is blocked through deletion of myomaker in SCs”) when these data are first discussed. They do this well in the Discussion, but it should be noted at the time the data are presented as well.*

We included this statement in the revised version (subsection “Fusion during MOV is blocked through deletion of myomaker in SCs”, first paragraph).